# CRISPR-Cas12a induced DNA double-strand breaks are repaired by multiple pathways with different mutation profiles in *Magnaporthe oryzae*

Jun Huang [1], David Rowe[1], Pratima Subedi[1], Wei Zhang[1], Tyler Suelter[1], Barbara Valent [1] & David E. Cook [1] ✉

CRISPR-Cas mediated genome engineering has revolutionized functional genomics. However, understanding of DNA repair following Cas-mediated DNA cleavage remains incomplete. Using Cas12a ribonucleoprotein genome editing in the fungal pathogen, *Magnaporthe oryzae*, we detail non-canonical DNA repair outcomes from hundreds of transformants. Sanger and nanopore sequencing analysis reveals significant variation in DNA repair profiles, ranging from small INDELs to kilobase size deletions and insertions. Furthermore, we find the frequency of DNA repair outcomes varies between loci. The results are not specific to the Cas-nuclease or selection procedure. Through Ku80 deletion analysis, a key protein required for canonical non-homologous end joining, we demonstrate activity of an alternative end joining mechanism that creates larger DNA deletions, and uses longer microhomology compared to C-NHEJ. Together, our results suggest preferential DNA repair pathway activity in the genome that can create different mutation profiles following repair, which could create biased genome variation and impact genome engineering and genome evolution.

The CRISPR-Cas9 (clustered regularly interspaced short palindromic repeats and CRISPR associated protein 9) genome editing platform has been widely used in multiple organisms including animals, plants, and fungi for functional genomics studies[1–6]. The basic requirement for CRISPR-Cas genome engineering is a Cas endonuclease protein complexed with a single-guide RNA targeting a genomic region following a protospacer adjacent motif (PAM), such as NGG in the case of the commonly used SpCas9 protein from *Streptococcus pyogenes*[1,2,7]. Another Cas effector, termed Cas12a (formerly named as Cpf1), is an alternative genome editing tool that has several unique features compared to Cas9 based effectors[8–14]. For instance, Cas12a recognizes a T-rich PAM, which can be better suited for editing some genomic regions[9]. Also, the RNase activity of Cas12a can process an array (single RNA molecule with multiple guide sequences) into multiple RNA

molecules of single sequences, which allows more convenient multiplex genome engineering[8]. The nuclease activity of Cas12a generates staggered DNA breaks, resulting in double-stranded DNA with 5' overhangs, compared to blunt-end double-stranded DNA following Cas9 nuclease activity[9].

A critical component determining the outcome of genome engineering is DNA double-strand break (DSB) repair, mediated by endogenous DNA repair machinery[15]. Proper repair of DNA DSBs, whether induced by Cas effectors or under natural conditions, is critical to maintain genomic stability, where repair failure can result in altered genome function and be potentially lethal[16–18]. DNA DSB repair is mediated by two major pathways, canonical non-homologous end joining (C-NHEJ) and homology directed repair (HDR)[19–21]. One of the major differences between

[1]Department of Plant Pathology, Kansas State University, Manhattan, KS 66506-5502, USA. ✉e-mail: decook@ksu.edu

C-NHEJ and HDR is the initial processing of DNA ends at a break site, where HDR requires extensive DNA end resection (i.e., enzymatic nucleotide removal from DSB site), which is inhibited by C-NHEJ[21,22]. In the initial steps of C-NHEJ, the Ku70-Ku80 heterodimer interacts with broken DNA ends to inhibit resection[23], and recruits additional proteins to the site eventually repairing the DSB via DNA ligase IV[19,24,25]. The C-NHEJ pathway does not rely on a homologous DNA template for repair, and commonly results in small insertions and deletions (INDELs)[19,26], but there are also examples of accurate C-NHEJ repair with and without DNA templates[27,28]. For DNA DSB repair via the HDR pathway, template DNA with extended homologous sequences (typically >100 bp) are used for what is generally considered accurate repair[29]. Two additional DNA DSB repair pathways, which also require end resection at DSB sites, are termed alternative end joining (a-EJ), and single strand annealing (SSA)[19,20,30]. The a-EJ pathway is also referred to as microhomology-mediated end joining (MMEJ), theta-mediated end-joining (TMEJ), and has been called alternative NHEJ (A-NHEJ) depending on the system and report[31–33]. While the three pathways involving end resection rely on homologous sequence for DSB repair, the length of homologous sequence used by a-EJ, SSA and HDR is different. The a-EJ repair pathway involves annealing microhomologous sequences (typically 2–20 bp) and gap filling by DNA polymerase theta (Polθ) near the DSB[34], resulting in small insertions, deletions and templated insertions in mammalian and plant systems[19,35,36]. The SSA pathway involves annealing with longer homologous sequences (>25 bp), often described to reside at longer distances from the DSB site and result in larger deletions as the result of removing 3′ non-homologous ssDNA via Rad1-Rad10 endonuclease[19,20,30,34].

Many questions remain for how the individual DNA repair pathways interact, such as their individual contributions to genome stability, their hierarchy for DSB repair, and variation in DSB repair pathways in microbial eukaryotes[33]. There have been conflicting reports on the importance and role of a-EJ for repairing DSBs[19,37], however, clear evidence shows that a-EJ substantially contributes to DNA repair in zebrafish embryos[38], mouse cell lines[39,40], *Caenorhabditis elegans*[41], and the model plant *Arabidopsis thaliana*[42]. Interestingly, while the genetic identification of C-NHEJ independent DSB repair was first described in *Saccharomyces cerevisiae*[43], there are no reports of TMEJ in yeast or filamentous fungi. There are also clear differences for DSB repair across fungi, such as HDR being highly active in yeast *S. cerevisiae*, while C-NHEJ predominates DNA DSB repair in most filamentous fungi[44–46].

In this research, we developed efficient genome editing using Cas12a-based ribonucleoprotein (RNP) in *Magnaporthe oryzae* (synonym of *Pyricularia oryzae*), a filamentous fungal pathogen of monocots threating world food security[47–49]. The use of CRISPR editing in fungi can increase the speed and efficiency of traditional gene replacement strategies, and the use of RNPs in *M. oryzae* can alleviate problems related to cytotoxicity and off-target mutations[50,51]. Surprisingly, we found that Cas12a editing in *M. oryzae* resulted in numerous mutants that contained severe DNA alterations at the targeted locus. Using long-read DNA sequencing and de-novo assembly, we confirmed at nucleotide resolution, multiple classes of DNA mutations, suggesting the involvement of different DNA repair mechanisms. The frequency of DNA repair outcomes after Cas12a editing were found to be locus-dependent across five tested loci. Similar severe DNA alternations were also observed with Cas9 edited mutants. *KU80* gene deletion confirmed the existence of both C-NHEJ-dependent and -independent pathways in *M. oryzae*. These results provide a detailed report of variable DNA repair outcomes after Cas12a-RNP editing, which have significant implications for natural and induced DNA repair in the *M. oryzae* genome.

## Results

### Cas12a ribonucleoprotein editing causes unexpected DNA mutations

To test Cas12a RNP gene editing in *M. oryzae*, we designed two gRNA targeting the *BUF1* locus that codes for a trihydroxynaphthalene reductase required for fungal melanin biosynthesis[52]. A DNA nuclease competent RNP comprised of purified LbCas12a protein (*Lachnospiraceae bacterium ND2006*) and *BUF1*-gRNA1 or -gRNA2 were transferred with donor DNA coding for the hygromycin resistance gene (*HYG*) into *M. oryzae* field isolate O-137 using protoplast transformation (Fig. 1a)[53]. The donor DNA contained short (30 and 35 bp) flanking sequences at the ends, homologous to the *BUF1* locus, to direct microhomology-mediated end joining (MMEJ donor DNA integration following Cas12a DNA DSB) (Fig. 1b). Transformed protoplasts were recovered on hygromycin selection, and hygromycin resistant (HYG^R) colonies were subsequently transferred to non-selective OTA plates to test for altered mycelial pigmentation. Across the experiments using both gRNAs, the *BUF1* locus was edited at a rate of ~77% (81 buff mutants/105 total HYG^R transformants). Among the 48 HYG^R transformants from *BUF1*-gRNA1, 43 of them displayed buff phenotype, and 38 of the 57 HYG^R transformants from *BUF1*-gRNA2 showed the buff phenotype (Fig. 1c, Supplementary Fig. 1 and Supplementary Table 1). All 32 control transformants transformed with donor DNA alone exhibited wild-type hyphal pigmentation (Supplementary Fig. 2 and Supplementary Table 1). To genotype and confirm Cas12a-mediated editing of *BUF1*, PCR was used to discriminate wild-type (~1.5 kb product) versus one-copy *HYG* donor insertion (~3.1 kb product) or other larger PCR product corresponding to the integration of *HYG* DNA donor, which we refer to as a 'simple insertion' (Fig. 1b, d). The Cas12a-mediated DSB could also be repaired to create an INDEL resulting in a PCR product indistinguishable from wild-type, while HYG^R transformants with wild-type pigmentation were presumed to have integrated the selectable marker at a secondary locus (Fig. 1d). Interestingly, ~93% of transformants (40/43) that displayed the mutant buff phenotype generated with *BUF1*-gRNA1, and almost ~87% generated with the *BUF1*-gRNA2 (33/38), failed to produce a *BUF1* PCR product (Fig. 1e, Supplementary Fig. 1 and Supplementary Table 1). The other 2/43 buff mutants generated with *BUF1*-gRNA1 and 4/38 buff mutants with *BUF1*-gRNA2 produced a clear PCR band consistent with simple insertion of the *HYG* coding sequence (Fig. 1e, Supplementary Fig. 1 and Supplementary Table 1). An INDEL mutation was detected in 1/43 and 1/38 buff mutants from *BUF1*-gRNA1 and *BUF1*-gRNA2 respectively, which displayed the wild-type sized PCR amplicon (Fig. 1e, f, Supplementary Fig. 1 and Supplementary Table 1). All recovered transformants that displayed a wild-type hyphal color produced the anticipated wild-type sized PCR product (Fig. 1e, Supplementary Fig. 1). The *BUF1*-gRNA1 targeted an intron and could therefore have generated DNA mutants that failed to produce a visible phenotype. To assess this, transformants generated with *BUF1*-gRNA1 that had a wild-type hyphal color and wild-type PCR amplicon were Sanger sequenced, which showed that all strains with normal pigmentation except for one transformant from replication two had wild-type *BUF1* sequence (Fig. 1e, f, Supplementary Fig. 1 and Supplementary Table 1). In order to test if the unexpected negative PCR from buff mutants was isolate-specific, we repeated the experiments in a different rice blast field isolate, termed Guy11[47]. Using *BUF1*-gRNA1, we found 30 HYG^R transformants all showed the buff phenotype. For *BUF1*-gRNA2, we recovered 32 HYG^R transformants, of which 30 had a buff phenotype. Further PCR genotyping these transformants showed that 27/30 buff mutants (*BUF1*-gRNA1) and 26/30 buff mutants (*BUF1*-gRNA2) failed to produce a PCR product from the *BUF1* locus (Fig. 1g, Supplementary Fig. 3 and Supplementary Table 1). Therefore,

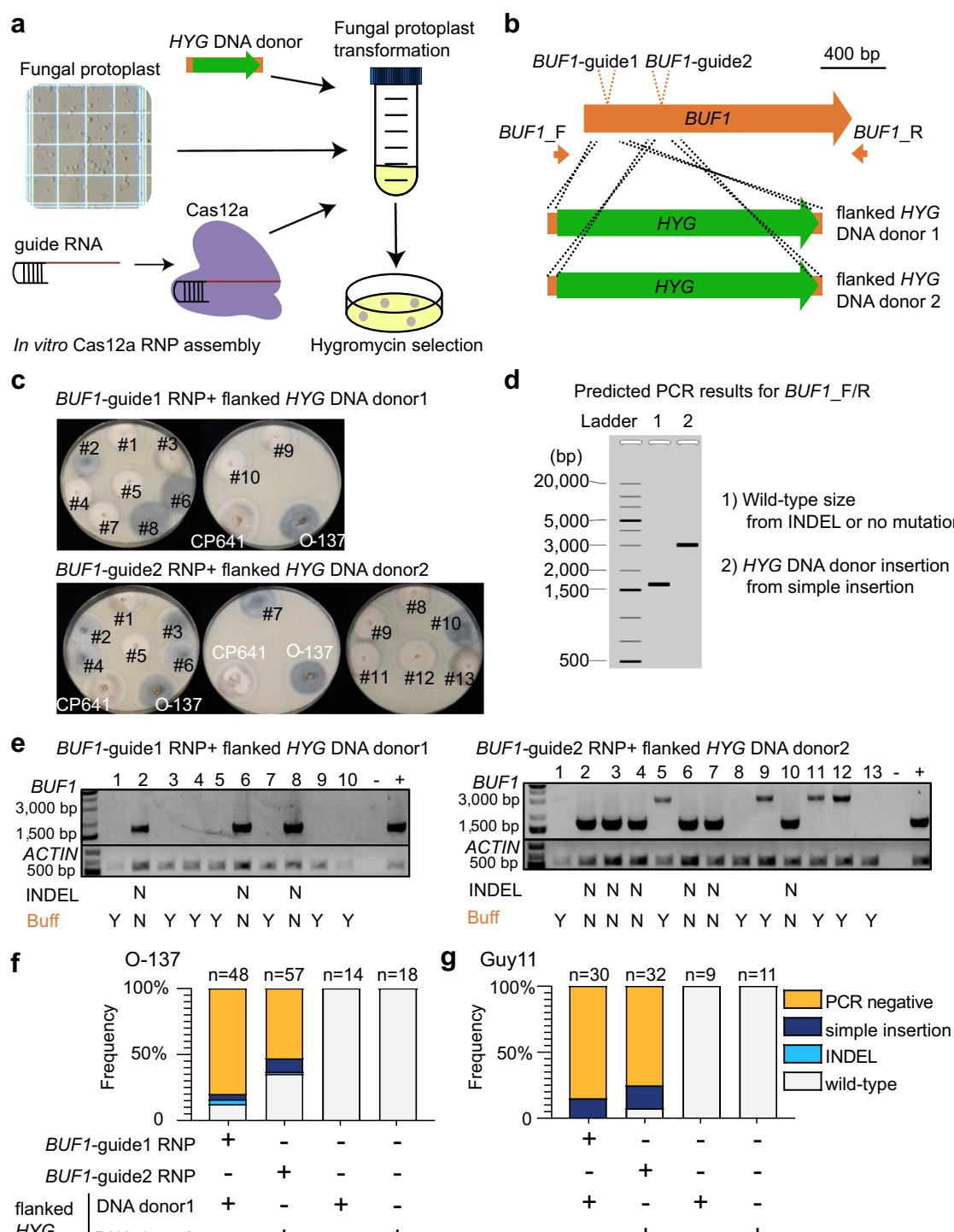

**Fig. 1 | Cas12a RNP mediated editing results in unexpected genotyping at the *BUF1* locus. a** Schematic diagram of CRISPR-Cas12a RNP mediated genome editing through protoplast transformation in *M. oryzae.* **b** Illustration of two *BUF1* (MGG_02252, 70-15 MG8 annotation) gRNAs design. Green rectangles indicate two different *HYG* DNA donors with flanked sequence homologous to the *BUF1* locus shown in orange. The location of PCR primer pair used for genotyping is shown (*BUF1*-F/R). **c** Phenotypes of hygromycin resistant (HYGᴿ) transformants plated on OTA. The wild-type O-137 is shown (dark grey hyphae) and a previously char-acterized Δ*buf1* in O-137 (CP641) showing the buff phenotype. Individual trans-formed colonies showing wild-type and buff color hyphae are shown labeled with numbers. **d** Diagram of expected results following PCR amplification from trans-formants. Ladder indicates the molecular weight ladder to determine product size, lane 1 (1) shows the expected size product for wild-type or small INDEL *BUF1* amplification, and lane 2 (2) shows the expected size product for *BUF1* amplification

where a single or multiple copy(s) of the *HYG* donor was inserted. **e** Genotyping results for the strains presented in the (**c**), the wild-type like PCR products from *BUF1* locus were purified and Sanger sequenced to detect potential INDELs. INDEL N indicates there were no INDELs observed after sequencing. Buff Y indicates the strain displayed the buff mutant color, while Buff N indicates wild-type phenotype. Lane (−) indicates negative control (water) and (+) a positive control O-137 genomic DNA used for PCR amplification. A separate PCR to amplify a portion of *ACTIN* was used as a DNA extraction control. The assay was repeated three times indepen-dently with similar results. **f**, **g** The frequency summary of DNA DSB repair out-comes in O-137 and Guy11. The number of independent PCR validated transformants (x) is listed (*n* = x) from three independent replications for O-137, and two independent replications for Guy11. The frequencies show the average outcome across replicates. Source data are provided as a Source Data file.

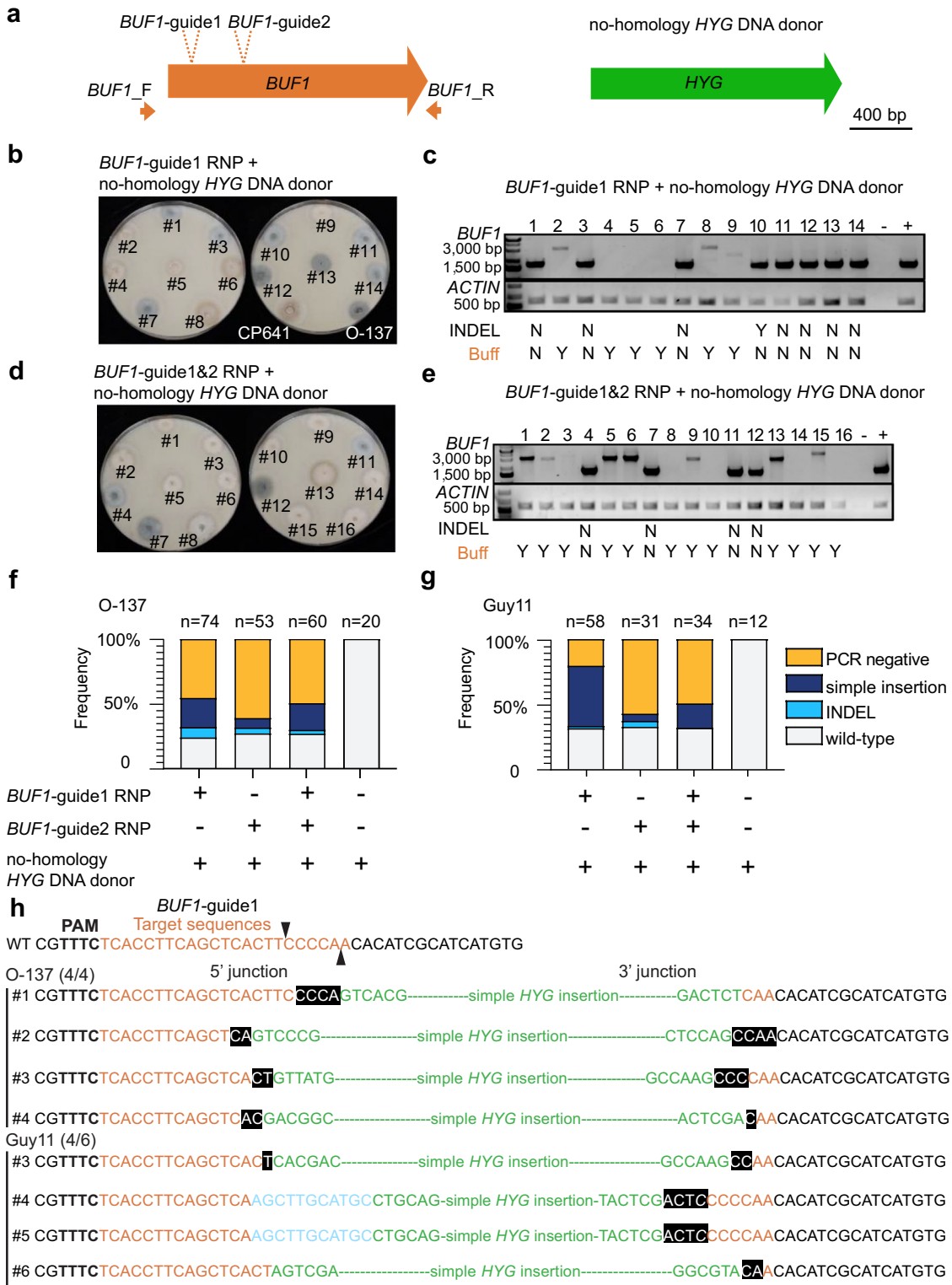

the results are consistent between the two strains, and indicate the observations are isolate-independent.

We had anticipated that the majority of mutants would have a simple insertion mediated by the homologous sequence on the donor DNA, but this only occurred in a total of ~9% of the O-137 and Guy11 *Δbuf1* mutants (13 *Δbuf1* mutants with 'simple insertion'/142 *Δbuf1* mutants) (Supplementary Table 1). Furthermore, the majority (~89%, 126 *Δbuf1* mutants with PCR negative/142 *Δbuf1* mutants) of repair events resulted in the inability to generate a PCR amplicon, suggesting a severe DNA alteration (Supplementary Table 1). From these

experiments, of the 142 *Δbuf1* mutants, we only recovered three *Δbuf1* (~2%) INDEL mutations (Fig. 1f, g and Supplementary Table 1).

## Microhomology can mediate donor DNA integration

Given that a majority of our *Δbuf1* mutants did not have a simple donor DNA insertion, we hypothesized that donor DNA lacking extended genome homology can be used to repair Cas12a-mediated DSB. That is, if the ~30 bp of homologous sequence had directed efficient integration of the donor DNA, we would have observed a higher frequency of simple insertions. To test this hypothesis, we again targeted the *BUF1*

**Fig. 2 | Microhomologous sequences are associated with donor DNA integrates at Cas12a target site. a** No-homology *HYG* DNA donor with *BUF1*-guide1 or/and guide2 RNP were used for protoplast transformation. **b, d** Phenotyping result of the HYG^R transformants from single or dual *BUF1* RNP targeting assays. The strains were plated on OTA for phenotyping. CP641 is a positive control (*Δbuf1*) for buff color hyphae, O-137 is the wild-type isolate used in the experiment. Individual transformed colonies showing wild-type and buff color hyphae are shown labeled with numbers. **c, e** Genotyping results for the strains presented in (**b** and **d**), the wild-type like PCR products from *BUF1* locus were purified and Sanger sequenced to detect potential INDELs. Image labels are the same as described for Fig. 1. The assay was repeated four (**c**) and three times (**e**) independently with similar results. **f** The frequency of DSB repair outcomes in O-137. The number of independent PCR validated transformants (x) is listed (*n* = x). The *BUF1*-guide1 was used for four independent transformations, the other transformations were repeated independently three times. The frequencies show the average outcome across replicates.

**g** The frequency of DSB repair outcomes in Guy11. The number of independent PCR validated transformants (x) is listed (*n* = x). The *BUF1*-guide1 was used for three independent transformations, the other transformations were repeated independently two times. The frequencies show the average outcome across replicates. Source data are provided as a Source Data file. **h** DNA sequence at the integration junction of simple insertion mutants from *BUF1*-guide1. The ratio to the right of the strain name is the number of mutants with at least 1 bp microhomology at the integration junction from the randomly selected simple insertion mutants Sanger sequenced. Bold letters indicate PAM sequences; orange letters indicate target sequences; black triangles highlight the potential Cas12a cut site. Green sequences are from *HYG* DNA donor; white letters in black boxes highlight microhomology (i.e., shared sequence) between the *BUF1* locus and donor DNA; blue letters are sequence insertions of unknown source. Italicized letters indicate SNP.

locus using Cas12a-RNP and the two gRNAs, but supplied a *HYG* donor DNA that lacked flanking sequences homologous to the *BUF1* locus (no-homology *HYG*) (Fig. 2a). Transforming strain O-137, we obtained 74 and 53 HYG^R transformants, from *BUF1*-gRNA1 and *BUF1*-gRNA2, respectively. We could confirm 58 of 74 (~78% for *BUF1*-gRNA1) and 44 of 53 (~83% for *BUF1*-gRNA2) HYG^R transformants had a DNA mutation at *BUF1*. When we considered these two guides together, we found that ~73% (74/102) of the edited strains were buff colored and produced no PCR product; ~19% (19/102) were buff color and had a simple insertion of DNA; ~6% (6/102) were wild-type color but had an intron INDEL from gRNA1; and ~3% (3/102) contained an INDEL from gRNA2 (Fig. 2b, c, f, Supplementary Figs. 4, 5, 7a, b and Supplementary Table 2). We additionally transformed strain O-137 with the two RNPs simultaneously (*BUF1*-gRNA1 and *BUF1*-gRNA2) and the no-homology *HYG* DNA donor (Fig. 2d), which resulted in a ~73% editing frequency (44 *Δbuf1* mutants/60 HYG^R transformants), where ~70% (31/44) produced no PCR product; 25% (11/44) had a simple insertion; and ~5% (2/44) had INDEL mutations (Fig. 2d, e, f, Supplementary Figs. 6a, b, 7c and Supplementary Table 2). To test whether the above observation is isolate-specific, both gRNAs were used to repeat the experiments in the Guy11 strain producing a ~72% editing efficiency (64 *Δbuf1* mutants/89 total HYG^R transformants), where ~52% (33/64) produced no PCR product; ~44% (28/64) had a simple insertion; and three of the 89 HYG^R transformants had an INDEL at *BUF1* (Fig. 2g, Supplementary Fig. 8, 10 and Supplementary Table 2). For the dual RNP transformation ~76% (26 *Δbuf1* mutants/34 HYG^R transformants) had a DNA mutation, where ~81% (21/26) showed the PCR negative genotype and ~19% (5/26) had a simple insertion (Fig. 2g and Supplementary Fig. 9a, b and Supplementary Table 2). Through Sanger sequencing randomly selected simple insertion transformants generated with *BUF1*-gRNA1, we found frequent ~2 bp microhomology (MH) between no-homology *HYG* DNA donor and *BUF1* locus at the integration junction (Fig. 2h). Additionally, to rule out that our observations are dependent on the *HYG* donor DNA and hygromycin selection, we performed the same experiments with a different donor DNA sequence coding for resistance to the drug G418 (Geneticin). We again recovered ~44% (32/72) G418 resistant (G418^R) transformants with the buff mutant phenotype and found that ~72% (23/32) were PCR negative and ~22% (7/32) had PCR amplification that suggested a simple insertion. We also recovered two transformants (2/32) that carried a *BUF1* INDEL from guide2 through tandem duplication or INDEL (Supplementary Fig. 11a, b, c, d, g, h and Supplementary Table 2).

From these experiments using a donor DNA with no-homologous sequence, we conclude that (i) donor DNA does not require extended homologous sequence to resolve Cas12a-mediated DSB; (ii) non-homologous donor DNA can integrate at DSB sites at a reasonably high frequency (~26% across all experiments, 70 mutants with simple insertions/268 total *Δbuf1* mutants); (iii) more severe DNA alterations resulting in no PCR products are common (~68% across experiments,

182 mutants with PCR negative/268 total *Δbuf1* mutants); (iv) INDELS were not common from these experimental conditions (~6% across experiments, 16 mutants with indel/268 total *Δbuf1* mutants); (v) targeted DNA mutation at *BUF1* is dependent on the Cas12a RNP complex, as 70 transformants obtained using no-homology *HYG* or *G418* donor DNA alone (i.e., in the absence of Cas12a RNP) did not cause the buff phenotype and among 50/70 that were PCR tested, none showed distinguishable *BUF1* DNA mutations (Fig. 2f, g and Supplementary Fig. 6c, d, e, 9c, d, 11e, f, g and Supplementary Table 2).

## Long-read sequencing and de novo assembly resolve genotypes following Cas12a-mediated DSB repair

We sought to further understand what DNA mutation occurred in the roughly 70% of buff mutants that failed to produce a PCR product at the *BUF1* locus. Eight O-137 derived buff mutants from the Cas12a RNP and no-homology *HYG* DNA donor transformation were selected for high-molecular-weight DNA extraction, nanopore sequencing and de novo assembly (*Δbuf1#2, -#4, -#5, -#6* from rep 1 in Fig. 2c, *Δbuf1#10* from Fig. 2e, *Δbuf1#1, -#5, -#13* from rep 4 in Supplementary Fig. 4c). All eight sequenced strains displayed the mutant buff color, where seven produced no PCR product, and one transformant (*Δbuf1#2* Fig. 2c) produced the ~3.1 kb PCR product we inferred to be a simple insertion of donor DNA. The eight strains were sequenced to an average depth of 52x and yielded highly contiguous assemblies (average N50 of 3.29 Mb) (Supplementary Table 3). These assemblies allowed for base pair resolution interrogation of *BUF1* DNA alterations. Consistent with PCR genotyping, the transformant thought to have a simple donor DNA insertion (*Δbuf1#2* Fig. 2c) indeed had an almost full copy of the hygromycin coding sequence (1328 bp) plus an additional hygromycin fragment (140 bp) at the Cas12a cut site (16 bp after the PAM sequence) based on the long-read assembly (Fig. 3a, mutant 1). The insertion was nearly scar free, with the junction sequence showing 2 and 3 bp of microhomology at the 5' and 3' ends respectively, and only one base pair was deleted at the Cas12a endonuclease site (Supplementary Fig. 12). The results from the other seven assemblies were grouped into one of three categories, namely, large insertion, large deletion, and deletion plus insertion (Fig. 3b, c, d). Two mutants (*Δbuf1#5, -#6* from rep 1 in Fig. 2c) each contained large insertions of concatemer *HYG* donor sequence, including promoter, coding, and terminator sequences, totaling 10 and 17 kb insertions respectively (Fig. 3b, mutant 2, 3). Not all the *HYG* DNA fragments were intact, and the coding sequences were in both the forward and reverse orientation (Fig. 3b). The insertion junction for both large insertion mutations had 2 bp of microhomology at the 5' end, and no homology at the 3' end, where in one mutant there was an error-free insertion at the locus, while the other mutant had a 17 bp deletion at the 3' end (Supplementary Fig. 12).

Assemblies from two *Δbuf1* mutants (Rep1-*Δbuf1#4* and Rep4-*Δbuf1#5*) identified the same ~21 kb deletion around the *BUF1* locus,

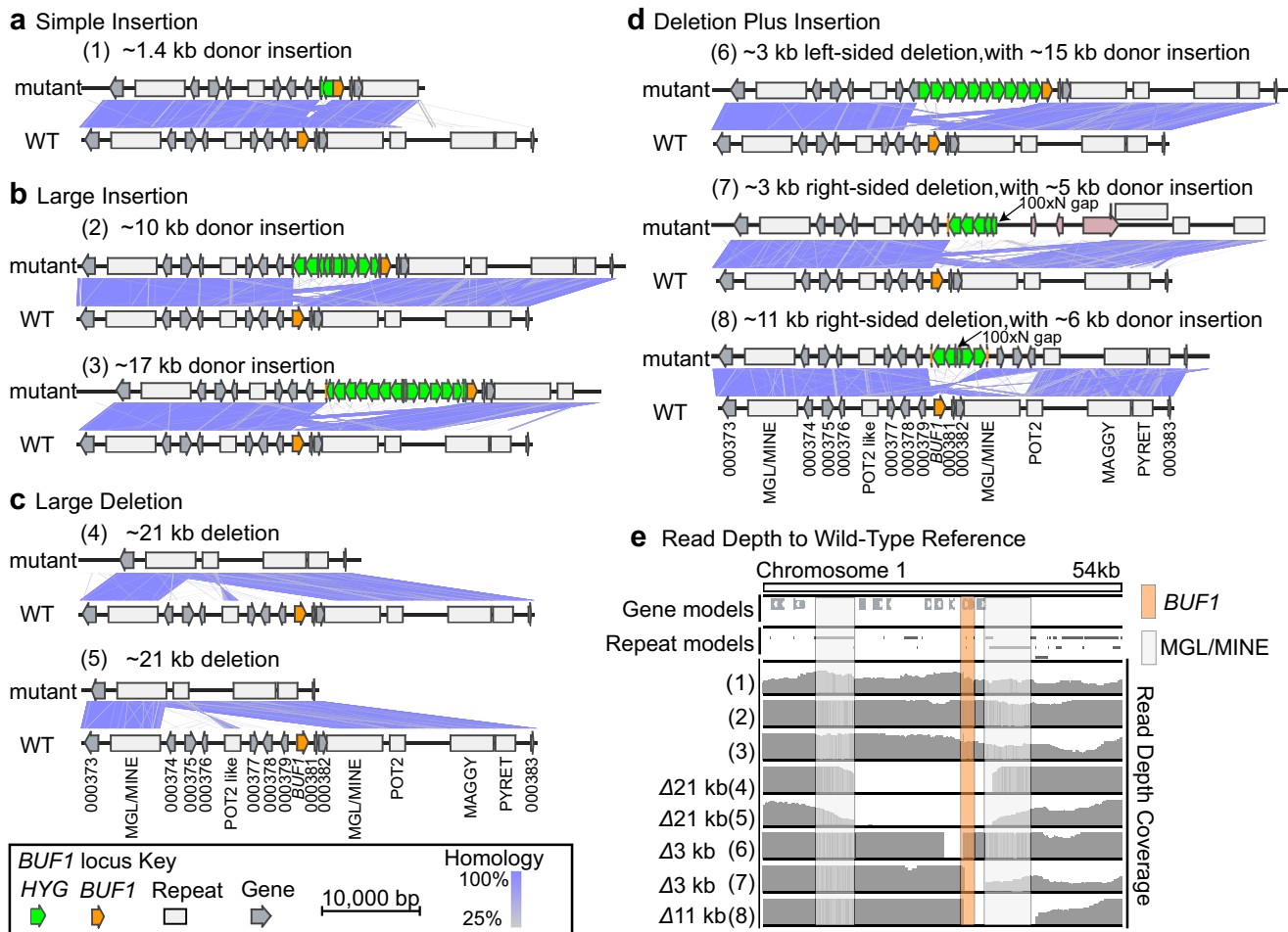

**Fig. 3 | Long-read assemblies reveal three non-canonical DNA mutation profiles following DSB repair.** Microsynteny between assembled edited strains (top) and the wild-type (bottom) indicated by purple connecting bands ranging from 25 to 100% similarity as indicated by the color scale. Individual assemblies were classified into one of four mutation classes (**a**–**d**): (**a**) simple donor insertion; (**b**) large donor DNA insertion; (**c**) large DNA deletion; and (**d**) DNA deletion plus donor insertion, where two assemblies were incomplete and required the identification of another contig to complete the locus. The merged region is indicated with an arrow and labeled 100xN gap. The pink labelled genes in (7) indicate three genes inserted from a *trans* locus. **e** Coverage of nanopore long-read mapping to the *BUF1* locus.

Reads were mapped to the O-137 wild-type genome, where reads were filtered to remove low quality mapping (MAPQ < 60). The *BUF1* gene is highlighted with as a vertical orange box while the flanking repetitive DNA (MGL/MINE) are highlighted with a light grey box. Symbols are shown in the key to indicate the *BUF1* coding sequence (orange arrow), *HYG* coding sequence (green arrow), other coding sequences (grey arrow), and annotated repetitive DNA (grey box). Mutants labeled (1), (2), (3), (4), (5), (6), (7) and (8) indicate Rep1-*Δbuf1#2*, Rep1-*Δbuf1#5*, Rep1-*Δbuf1#6*, Rep1-*Δbuf1#4*, Rep4-*Δbuf1#5*, Rep4-*Δbuf1#1*, Rep1-*Δbuf1#10* and Rep4-*Δbuf1#13* mutants respectively (Supplementary Table 3).

where *BUF1* and eight additional genes were all deleted (Fig. 3c, mutant 4, 5). Interestingly, the assemblies indicated the deletions took place between similar flanking non-LTR retrotransposons, which appear to be nested insertions of a LINE element, termed MGL, inserted into a hybrid LINE element termed MINE[54,55] (Supplementary Fig. 13). The deletions suggest that homology between the two elements was used to resolve the break, potentially through the SSA pathway, resulting in a single retrotransposon copy and the 21 kb deletion (Fig. 3c, mutant 4, 5, Supplementary Fig. 13). The *HYG* coding sequence for these two mutants was identified at independent loci on other chromosomes (Supplementary Fig. 14a, b). In mutant 4, the *HYG* insertion was a large concatemer of ~20 kb, while the other deletion mutant had two *HYG* copies inserted (Supplementary Fig. 14a, b). To confirm the assembly-based deletion, a ~6.7 kb PCR product that spanned the break resolution junction (MGL/MINE) was amplified in mutant 4, which failed to amplify a product in the wild-type as expected (Supplementary Fig. 15). The remaining three mutants (Rep4-*Δbuf1#1*, Rep1-*Δbuf1#10* and Rep4-*Δbuf1#13*) had both *BUF1* locus deletions, ranging in size from 3 to 11 kb on either the 5' or 3' side of the Cas12a endonuclease site, along

with large insertions of concatemer donor DNA (Fig. 3d, mutant 6,7,8). The assemblies for mutants 7 and 8 did not completely resolve the *BUF1* locus in a single assembled contig. Here, we identified two contigs in both mutant assemblies with sequence homology to the *BUF1* locus, which were joined and analyzed. Interestingly, these results indicated the insertion of additional genomic DNA from other regions of the genome, including the insertion of coding sequences (Fig. 3d). As noted for the other mutation types, we also observed microhomology (2–3 bp) at three of the four integration junction sites from mutants 7 and 8 (Supplementary Fig. 16).

In order to validate these mutation outcomes and support the assembly results, the long-reads from nanopore sequencing were mapped to the O-137 reference genome. If the *BUF1* mutation and deletion genotypes were the result of an assembly error, mapping the DNA sequence reads to the reference assembly would recover the *BUF1* DNA. For all five mutants where the assembly and original PCR indicated a deletion, there was no sequenced DNA mapping to the reference (Fig. 3e). Strains that contained *HYG* insertions, but no alteration to the native *BUF1* sequence showed uniform coverage

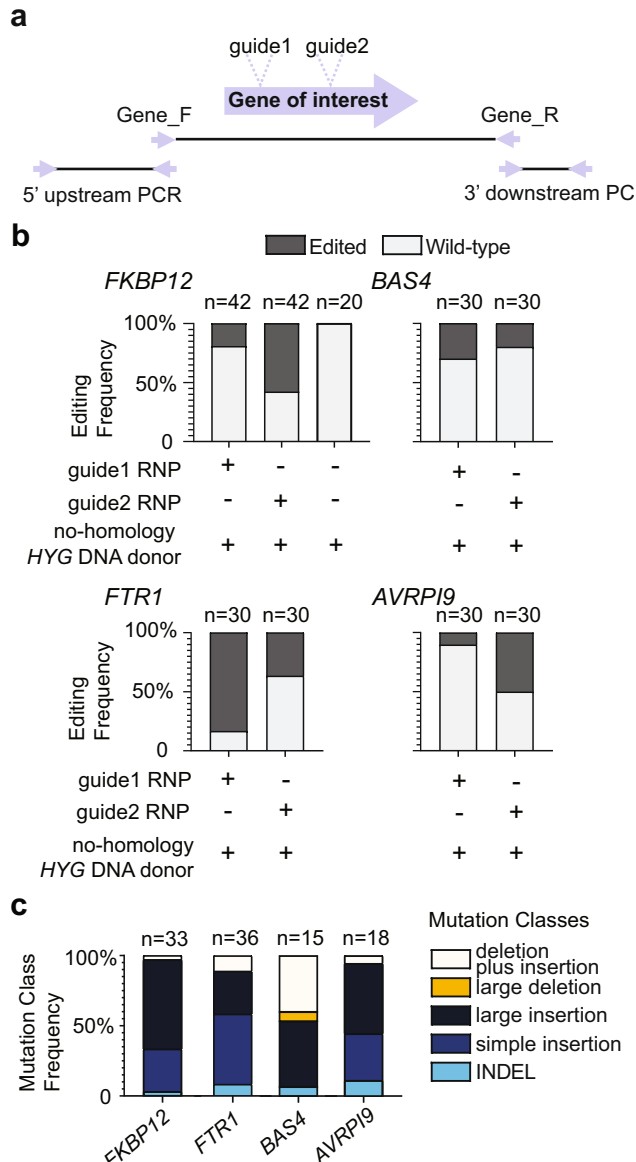

**a**

guide1 guide2

Gene of interest

Gene_F          Gene_R

5′ upstream PCR          3′ downstream PCR

**b**

Edited   Wild-type

*FKBP12*          *BAS4*

*FTR1*          *AVRPI9*

**c**

Mutation Classes

deletion plus insertion
large deletion
large insertion
simple insertion
INDEL

**Fig. 4 | DNA repair outcomes following Cas12a editing differ among multiple loci. a** Schematic illustration for CRISPR-Cas12a editing experiments. Two different guides were tested for each locus. Gene_F/R primer pairs were used for a first round of genotyping to determine the type of mutation present in a transformant. PCR negative strains were subsequently genotyped using the 5′ upstream and 3′ downstream primer pairs per gene to further genotype the occurrence of large-scale DNA alternations. **b** The editing frequencies at *FKBP12* (MGG_06035), *FTR1* (MGG_02158), *BAS4* (MGG_10914), and *AVRPI9* (MGG_12655) indicated by color (dark grey for mutant, light grey for wild-type). The MGG numbers based on the 70-15 assembly are provided for clarity. The number of independent transformants genotyped (x) is listed (*n* = x) collected from two independent transformations. The frequencies show the average outcome across replicates. **c** The frequency of mutations in each class at *FKBP12, FTR1, BAS4* and *AVRPI9* are shown, class designation is indicated by the color key to the right. The number of mutants determined from (**b**) used for genotyping (x) is listed (*n* = x). The class frequencies are reported as the total across all experiments. Source data are provided as a Source Data file.

across the locus. These results support that large deletion and deletion plus insertion mutants lost DNA corresponding to the *BUF1* locus consistent with the de novo assemblies.

Given that more than half of the identified buff mutants across our experiments failed to produce PCR products, we were interested to use the assembly identified genotypes (large insertion, large deletion,

and deletion plus insertion) to screen the previously PCR negative transformants. For this, we designed primer pairs to amplify small fragments at the 5′ upstream and 3′ downstream of the *BUF1* locus (Supplementary Fig. 17a). Our results showed that ~64% (67/105) of O-137 derived PCR negative transformants with no-homology *HYG* DNA donor and *BUF1* RNP had a large insertion (i.e., both 5′ and 3′ amplified PCR products); ~18% (19/105) had a large deletion (i.e., both 5′ and 3′ PCR failed) and ~18% (19/105) had a deletion plus insertion (i.e., PCR amplified product at either 5′ or 3′ end) (Supplementary Figs. 4, 5b, c, 6a, b, 17a, b, d and Supplementary Table 2). We confirmed these results by also genotyping the transformants generated using the Guy11 derived strain, where we found that ~81% (44/54) had a large insertion; ~7% (4/54) had a large deletion and ~11% (6/54) had a deletion plus insertion (Supplementary Fig. 8d, f, 9b, 17a, c, d and Supplementary Table 2).

**DNA repair outcomes following Cas12a editing differ between multiple loci**

The *BUF1* locus has been characterized as unstable[52,56], and we were interested to understand if the unexpected DSB repair outcomes found at *BUF1* were representative of other loci in the genome. Wild-type *M. oryzae* is sensitive to the drug FK506, while disruption of the corresponding receptor, *FKBP12*, causes insensitivity to FK506 (Supplementary Fig. 18)[57,58]. Therefore, we targeted *FKBP12* using Cas12a RNP and two separate gRNAs (*FKBP12*-guide1 and-guide2) and utilized sensitivity to FK506 to identify mutants (Fig. 4a). In order to compare the results to those from *BUF1, FKBP12* editing included the no-homology *HYG* DNA donor and hygromycin selection (Fig. 4b and Supplementary Table 4). We obtained 84 HYG[R] colonies, of which 33 (~39% editing efficiency) were insensitive to FK506 and presumably carried a mutation at the *FKBP12* locus, while none of the 20 no-homology *HYG* DNA alone transformants showed FK506 resistance (Fig. 4b, Supplementary Fig. 19 and Supplementary Table 4). The same PCR amplification strategy was used to genotype the FK506 insensitive mutants, and we found that ~30% (10/33) had a simple insertion; ~64% (21/33) had a large insertion; ~3% (1/33) had a deletion plus insertion and ~3% (1/33) contained INDEL (Fig. 4a, c, Supplementary Fig. 19a, b, c, d and Supplementary Table 4). The PCR genotyping indicated that none of the FK506 insensitive strains had large deletions (Fig. 4a, c, Supplementary Fig. 19a, b, c and Supplementary Table 4).

We additionally tested three other loci, one coding for an annotated plasma membrane iron permease (*FTR1*), which is 50 kb away from the *BUF1* locus, an apoplastic secreted protein *BAS4*, and an avirulence protein *AVRPI9*, both of which are presumed to help *M. oryzae* facilitate host infection[59–61]. Two independent gRNAs were designed for each of the three loci, which were transformed as Cas12a-RNPs with the no-homology *HYG* DNA donor (Fig. 4b). From editing *FTR1*, 60 HYG[R] transformants were selected and genotyped. This gene had a high editing efficiency at 60% (36/60), and of the strains carrying a mutation, we found that ~8% (3/36) had an INDEL; 50% (18/36) had a simple insertion; ~31% (11/36) had a large insertion; ~11% (4/36) had a deletion plus insertion and no large deletions were recovered (Fig. 4b, c, Supplementary Fig. 20 and Supplementary Table 4). The editing efficiency at *BAS4* was 25% (15 Δ*bas4* mutants/60 HYG[R] transformants), where ~6% (1/15) of the mutants had an INDEL; ~46% (7/15) had a large insertion; ~6% (1/15) resulted from a large deletion; 40% (6/15) from a deletion plus insertion (Fig. 4b, c, Supplementary Fig. 21 and Supplementary Table 4). Interestingly, no simple insertions were recovered from the *BAS4* transformants using either of the two guides, despite simple insertion mutants being commonly found for the *BUF1* (19%), *FKBP12* (30%), and *FTR1* (50%) loci. Also of note, *BAS4* editing resulted in near half of the identified mutants having a deletion plus insertion or large deletion, which was not observed for other tested loci. The editing efficiency at the *AVRPI9* locus was ~30% (18 Δ*avrpi9* mutants/60 HYG[R] transformants), and of these 18 mutants, nine resulted from a

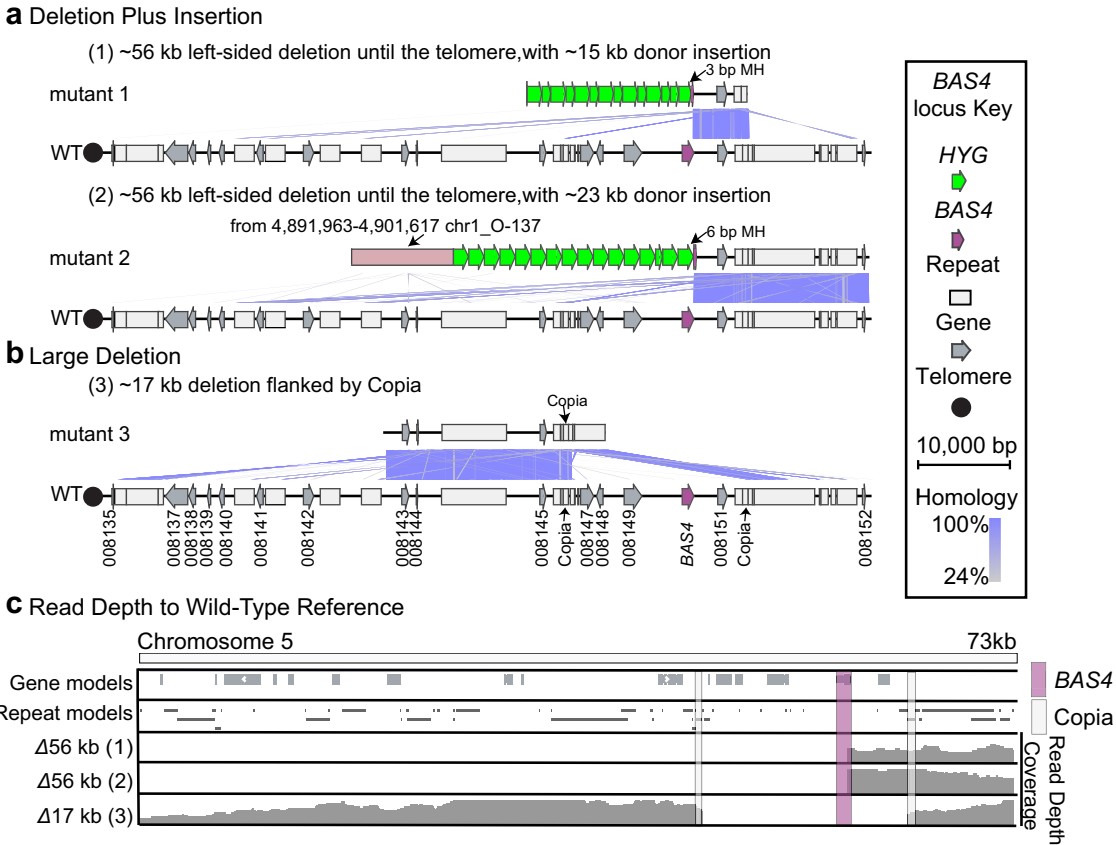

**Fig. 5 | Large-scale DNA deletions at subtelomeric *BAS4* locus.** Microsynteny between assembled *bas4* edited strains (top) and the wild-type (bottom) indicated by purple connecting bands ranging from 24 to 100% similarity as indicated by the color scale. Three *Δbas4* mutants with two mutations classes (**a**) deletion plus insertion and (**b**) large deletion were sequenced and displayed. The pink labelled segment in (2) indicated sequence inserted from different chromosome. MH indicates the shared microhomology. **c** Coverage of nanopore long-read mapping from telomere to the *BAS4* locus. Reads were mapped to the O-137 wild-type genome, where reads were filtered to remove low quality mapping (MAPQ < 60). The *BAS4* gene is highlighted with as a vertical purple box while the flanking repetitive DNA (Copia) are highlighted with a light grey box. Symbols are shown in the key to indicate the *BAS4* coding sequence (purple arrow), *HYG* coding sequence (green arrow), other coding sequences (grey arrow), annotated repetitive DNA (grey box) and telomere (black dot). Mutants labelled as (1), (2) and (3) indicate *Δbas4*#2, -#3 and -#11 mutants from rep1 in Supplementary Fig. 21a, respectively (Supplementary Table 3).

large insertion; six had a simple insertion; one contained deletion plus insertion and the other two lines carried INDELs (Fig. 4b, c and Supplementary Fig. 22). While we did not exhaustively test gRNA at all loci, the results using two independent gRNA at each locus suggests that editing efficiency is not the same across the genome, which has been reported in other organisms[62]. An unexpected and novel finding, however, was that the spectrum of mutations resulting from DSB repair did not occur at equal proportions for the tested loci. Indeed, formal testing of the different edited loci into the five classes of DNA mutations indicated that the highly significant association between the loci and DNA mutation outcomes, indicating a non-random association between repair mutation outcome and specific loci (Fisher's exact test, two-sided, *p*-value = 0.0009995).

Among the loci we tested, *BAS4* showed the most dramatic mutation pattern, with more than half the mutations being large-scale deletions (i.e., large deletion or deletion plus insertion). In order to investigate the potential mechanisms of deletion at *BAS4* locus and compare the detailed mutational profiles to *BUF1*, we nanopore long-read sequenced three additional transformants. We sequenced two strains that had deletion plus insertion PCR genotypes and one that had a large deletion PCR genotype (*Δbas4*#2, -#3 and -#11 from rep1 in Supplementary Fig. 21a, Supplementary Table 3). The assemblies were consistent with the PCR genotyping, and both insertion plus deletion contigs contained a portion of the *BAS4* locus and long concatemers of

the *HYG* insert DNA (Fig. 5a and Supplementary Fig. 21 and Supplementary Table 3). However, the contigs did not contain sequence to the 5' of the *BAS4* locus, and surprisingly, mapping the long-read sequences to the reference genome indicated that the transformants did not contain DNA all the way to the telomere, some 56 kb away from the *BAS4* targeting site (Fig. 5c). This was observed for both independent mutants termed 1 and 2 (Fig. 5c). For one of these mutants, ~9 kb of trans-chromosome genomic DNA was found in the *BAS4* locus assembly (Fig. 5a, mutant 2). As observed at the *BUF1* locus, 3 bp and 6 bp microhomology shared between *BAS4* locus and donor DNA was also found at the repair junction (Fig. 5a). For the large deletion mutant, the assembly and read coverage identified a ~17 kb deletion had occurred. Interestingly, the borders of the deletion are occupied by two identical LTR retrotransposon Copia elements (Fig. 5b, c). The large deletion of DNA occurring between flanking repetitive elements was therefore seen at both *BUF1* and *BAS4*. These results indicate that proximity to the telomere, as well as flanking repetitive DNA, could influence DNA DSB repair outcomes.

## Locus-dependent DNA mutation frequency still occurs under a different editing scheme
Our initial editing experiments strongly favored donor DNA integration at the Cas12a-edited locus because of the induced DSB and selection on hygromycin. Given that this method was used for all the

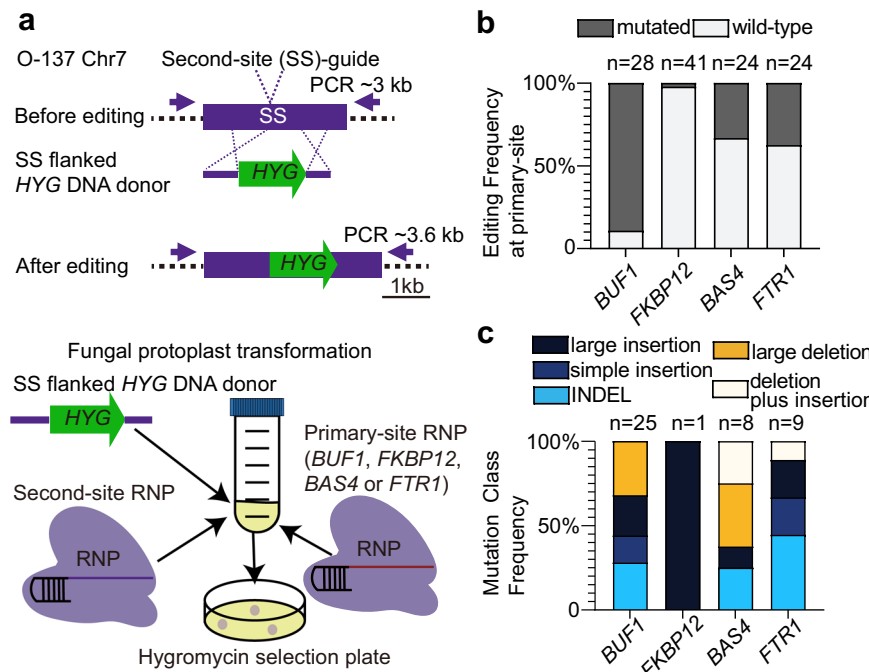

**Fig. 6 | Distinct DSB repair pattens following donor insertion at a second integration site. a** Schematic illustration of second-site (SS) insertion assay. The *HYG* coding sequence was directed to a second site through long homologous sequence located in a non-coding region on chromosome7 (chr7_O-137:1,521,898–1,524,866). Fungal protoplasts were simultaneously transformed with two RNPs, one targeting the second-site and the other targeting the locus of interest. **b** The editing frequencies for the four tested loci following *HYG* selection (dark grey for mutant, light grey for wild-type) shown as the average outcome across replicates. The number of independent PCR validated transformants (x) is listed (*n* = x). The guides were used for two independent transformations. **c** The frequency of distinct DNA repair outcomes for the four tested loci, colored according to the respective DNA mutation class. The class frequencies are reported as the total across all experiments. The number of mutants determined from (**b**) used for genotyping (x) is listed (*n* = x). Source data are provided as a Source Data file.

edited loci in this work, bias introduced by this experimental set-up should be similar among the loci, and not account for the observed differences. Nevertheless, to further show that the locus-dependent DNA mutation frequencies were not dependent on *HYG* insertion at the target locus, we devised a second editing scheme. Here, we developed an assay where the donor DNA for selection was targeted to a separate non-coding locus, termed second-site (SS), by transforming with two distinct RNPs at once (Fig. 6a). This approach should capture DSB repair outcomes for the locus of interest (i.e., primary-site), without the need for donor DNA integration at the target locus, thereby decoupling donor DNA integration and selection from DSB repair outcomes. To target the *HYG* donor DNA to the second site, we added long DNA sequence (730 bp and 518 bp) homologous to the second-site targeted site to the *HYG* donor DNA, which has been shown to increase homology directed repair, and thereby increase the frequency at which *HYG* inserted at the second-site and not the primary site of interest (Fig. 6a). Using this approach, we found high editing efficiency for the *BUF1* locus (89%, 25 *Δbuf1* mutants/28 HYG^R transformants) (Fig. 6b and Supplementary Table 5), which was similar to the earlier results obtained (~78% with *BUF1*-guide1). The proportion of the five types of DNA mutations at *BUF1* were 28% (7/25) INDELs; 16% (4/25) simple insertion; 24% (6/25) large insertion; 32% (8/25) large deletions; while no deletion plus insertions were detected (Fig. 6c, Supplementary Fig. 23 and Supplementary Table 5). These editing experiments returned substantially more INDEL mutations at *BUF1* than initially observed, consistent with the donor DNA integrating at the SS locus and not requiring it for DSB repair. Interestingly, the majority of recovered mutants still indicated either donor DNA insertion or large DNA deletion at the *BUF1* locus. The editing efficiency for the *FKBP12* locus was low, with only one mutant recovered from 41 transformants (Fig. 6b). The editing efficiency for this locus was also

low for the single RNP assay, but it was not clear why it was even lower for this SS editing scheme. The one *Δfkbp12* mutant contained a large insertion at the locus (Fig. 6c and Supplementary Fig. 24). For the *BAS4* locus, we obtained ~33% editing efficiency (8 *Δbas4* mutants/24 HYG^R transformants), of which, 25% (2/8) were INDELs; 12.5% (1/8) were large insertion; 37.5% (3/8) were large deletions; and 25% (2/8) were deletion plus insertions. (Fig. 6b, c, Supplementary Fig. 25 and Supplementary Table 5). Similar to initial single RNP assay, more than half of the *Δbas4* mutants were large deletion or deletion plus insertion mutations. The *FTR1* locus had a lower editing efficiency (9 *Δftr1* mutants/24 HYG^R transformants) for the second-site assay compared to the initial single RNP editing with gRNA1 (38% versus 83%, respectively). Despite the reduction in editing efficiency and increased INDELs, the proportion of DNA mutation outcomes was quite similar between the two editing schemes, where the second-site assay showed ~44% (4/9) INDELs; ~22% (2/9) simple insertions; ~22% (2/9) large insertions; and ~11% (1/9) deletion plus insertion mutations (Fig. 6b, c, Supplementary Fig. 26 and Supplementary Table 5). In neither experiment did we recover large deletion mutants at *FTR1*. We additionally genotyped the second-site for each mutant strain and found PCR negative results for the second-site in most of cases (Supplementary Fig. 23, 24, 25 and 26). This suggested that the addition of long homologous sequences did direct the donor DNA to the second-site, but it did not provide precise homology directed repair in the form of a single donor DNA insertion. As controls, the second-site was also edited with Cas12a SS-RNP and second-site flanked *HYG* DNA donor in the absence of the primary site RNP and similar PCR negative results were found (Supplementary Fig. 27a, b and Supplementary Table 5). Also, second-site flanked *HYG* donor DNA was transferred alone, without RNPs, and HYG^R transformants did not show the buff mutant color or have FK506 resistance, nor did the transformants produce aberrant PCR products when

amplifying the second-site locus, showing the mutations at these two primary sites, and the second-site were dependent on Cas12a-mediated DSB induction (Supplementary Fig. 27c, d and Supplementary Table 5). A limitation of this approach is how differences in RNP rates effect primary and secondary site editing. Differences between gRNA may have caused variation in the editing frequency, where large differences in primary and secondary site editing may have influenced the uptake of donor DNA. It is also possible that error-free editing took place at a higher rate for say the *FKBP12* locus, but we cannot determine that at this time. Overall, these results show that under a different editing scheme, which did not require donor DNA integration at the editing site of interest, we still observed that DNA mutation outcomes were not the same between the tested loci.

## The recovered spectrum of DNA mutations is not Cas12a dependent and also seen for Cas9 edited strains

Distinct DNA end structures, such as blunt ends caused by Cas9 versus overhang ends created by Cas12a, can influence DNA DSB repair pathway choice and mutational outcome[9,63,64]. Therefore, we tested if the range of DNA repair mutation profiles observed following Cas12a editing were also induced by the Cas9 nuclease. In order to reduce the bias caused by donor DNA integration, we edited the *FKBP12* coding sequence with either Cas12a RNP or Cas9 RNP followed by direct selection with the FK506 antibiotic (Fig. 7a). We designed a new Cas9 *FKBP12*-guide1 that is adjacent to the Cas12a *FKBP12*-guide1 from earlier assays, to aid comparison between the nucleases (Supplementary Table 7 and Fig. 7c). We obtained 51 and 50 FK506 resistant (FK506[R]) transformants from Cas12a and Cas9 editing respectively. Among them 1/51 FK506[R] transformants from Cas12a and 4/50 FK506[R] transformants from Cas9 showed PCR negative genotyping for the *FKBP12* locus (Fig. 7b, Supplementary Figs. 28a, 29a, 30a, 31a and Supplementary Table 6). The PCR positive transformants were Sanger sequenced to determine the mutation profiles generated by Cas12a and Cas9 editing, totaling 50 from Cas12a and 45 from Cas9. The mutations were categorized as (I) deletions with no MH, 24% (12/50) for Cas12a and ~29% (13/45) of Cas9 edited strains; (II) deletions with 1–2 bp MH, 64% (32/50) of Cas12a and ~16% (7/45) for Cas9 edited strains; (III) deletions with ≥3 bp MH, 2% (1/50) of Cas12a and ~11% (5/45) of Cas9 edited strains; (IV) insertions, 10% (5/50) of Cas12a and ~44% (20/45) of Cas9 edited strains displayed either minor DNA insertions from neighboring sequences (i.e., tandem duplications) or more complicated DNA insertions from other nuclear chromosomes or mitochondrial sequence (Fig. 7c, d and Supplementary Fig. 28b, 29b, 30b, 31b and Supplementary Table 6). We observed that the size of DNA mutations from PCR positive transformants were significant larger when Cas9 edited compared to those created with Cas12a (Mann Whitney test, two-sided, *p*-value = 0.0395) (Fig. 7c, e, Supplementary Figs. 28b, 29b, 30b and 31b). Overall, we found that both Cas12a and Cas9 induced DNA DSBs can result in the same classes of DNA repair mutations. Our results indicated that PCR negative mutations may occur more frequently with Cas9, and that Cas9 tended to cause more insertions, consistent with previous findings[63]. In *M. oryzae*, we observed a similar range of DNA mutation outcomes regardless of the nuclease used.

## C-NHEJ-independent repair in *M. oryzae* produces a different DNA mutation profile

The results from our experiments indirectly suggested that at least one additional a-EJ DNA repair pathway is active in *M. oryzae*, in addition to C-NHEJ and HR. To directly test this hypothesis, a required component of C-NHEJ, *KU80* (MGG_10157), was deleted. We generated the *Δku80* strain in the O-137 background through transferring two Cas12a targeting RNPs and a long-flanking *G418* DNA donor (>1 kb homology to the *KU80* locus) (Supplementary Fig. 32a). PCR genotyping result confirmed the deletion of *KU80* coding sequence and the correct

integration of *G418* DNA donor (Supplementary Fig. 32b). After single spore purification, we randomly picked one of the *Δku80* mutants (*Δku80#9* single spore8, hereinafter termed *Δku80*) for the further investigations. To confirm that Ku80 has canonical function in *M. oryzae*, we assessed the effect of *Δku80* on HR DNA DSB repair, where inhibition of C-NHEJ should promote HR efficiency[45,46]. Sequence homologous to *FKBP12* (>1 kb) was added to the *HYG* coding donor DNA and transformed into wild-type O-137 and *Δku80* (Supplementary Fig. 33a). The transformants that occurred via HR repair should be both hygromycin and FK506 resistant, while non-HR repair would cause only hygromycin resistance but not target *FKBP12* (i.e., FK506 sensitive). Indeed, we observed significantly higher *FKBP12* disruption for HYG[R] colonies when transforming the *Δku80* mutant compared to wild-type O-137 (97% versus 39% on average, unpaired *t*-test, two-sided, *p*-value = 0.0003) (Fig. 8a, Supplementary Fig. 33b, c). Thus, increased frequency of HR-mediated replacement confirmed the deficiency of C-NHEJ pathway in *Δku80* mutant line.

Next, we tested the dependency of C-NHEJ on our observed DNA mutation profiles following DSB repair. The *BUF1* locus was targeted with *BUF1*-guide2 RNP along with the no-homology *HYG* DNA donor for hygromycin selection. However, after four rounds of independent transformation, we did not recover any HYG[R] transformants in *Δku80*, while transformation in the wild-type O-137 background consistently yielded HYG[R] transformants. This suggested that C-NHEJ is required for the insertion of no-homology *HYG* DNA donor into the genome. Therefore, we could not assess DNA repair using this approach in *Δku80*. As an alternative, we targeted the *FKBP12* coding sequence with Cas12a using *FKBP12*-guide2, and directly selected mutations with FK506 selection, eliminating the need and bias of donor DNA uptake. Following editing and selection, we found ~33% (20/61) of FK506[R] transformants were PCR negative from *Δku80*, which is significantly higher than the frequency in wild-type O-137 (4%, 2/50, unpaired *t*-test, two-sided, *p*-value = 0.0356) (Fig. 8b, Supplementary Figs. 34a, 35a, 36a, 37a, b and Supplementary Table 6). Further amplifying 5' upstream and 3' downstream regions adjacent to *FKBP12* suggested that all 22 PCR negative mutants were the result of large-scale deletions (Supplementary Fig. 35a, 36a, 37a, b and Supplementary Table 6). The base-pair level mutation profile for PCR positive FK506[R] transformants was further assessed using Sanger sequencing. These results revealed two major findings. First, deletions in the *Δku80* background primarily contained ≥3 bp microhomology (MH) at the deletion junction (~98%, 40/41), which was significantly more than observed in the wild-type O-137 strain (~6%, 3/48) (unpaired *t*-test, two-sided, *p*-value < 0.0001) (Fig. 8c, d, Supplementary Fig. 34b, 35b, 36b, 37c and Supplementary Table 6). The remaining PCR positive mutant in the *Δku80* background (~2%, 1/41) contained a deletion with 1-2 bp MH. (Fig. 8d, Supplementary Fig. 36b and Supplementary Table 6). In the wild-type O-137 strain, PCR positive FK506[R] transformants contained ~42% (20/48) deletions without MH; ~38% (18/48) deletions with 1–2 bp MH; and ~15% (7/48) insertions (Fig. 8c, d, Supplementary Fig. 34b, 35b and Supplementary Table 6). The second major finding was that editing in *Δku80* resulted in significantly larger DNA sequence alternations compared to wild-type, +2 bp insertion versus −36 bp deletion on average, respectively (Mann Whitney test, two-sided, *p*-value < 0.0001) (Fig. 8e). These findings are consistent with the expected role of Ku80 in DNA end protection, and the proposed more mutagenic outcomes of MMEJ. From these experiments we conclude that the large DNA deletion mutations we observed are caused by C-NHEJ-independent DSB repair pathway(s), and that these large deletions are likely suppressed by Ku80-mediated DNA end protection in the wild-type. Small DNA deletions, on the order of 21–245 bp that contain ≥3 bp microhomology at the repair junctions are likely caused by MMEJ, while smaller deletions, on the order of 1–9 bp that contain no or 1–2 bp microhomology are likely repaired by the C-NHEJ pathway.

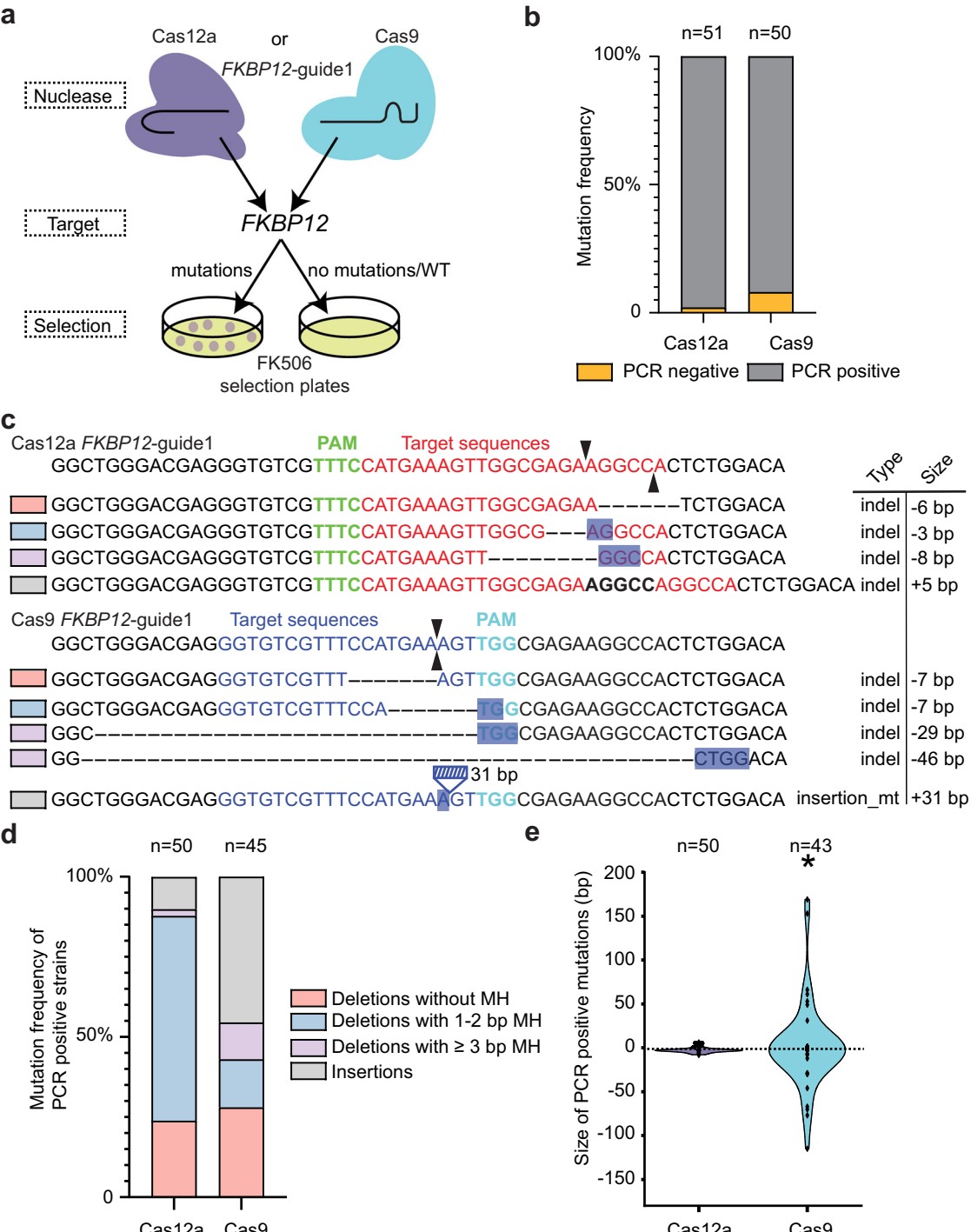

**Fig. 7 | Distinct mutational profiles caused by Cas12a and Cas9 in the absence of a DNA donor. a** Schematic illustration of a new editing scheme targeting *FKBP12* locus by using either Cas12a or Cas9 *FKBP12*-guide1 RNP with FK506 direct selection. **b** The mutation frequency of PCR negative and PCR positive outcome in Cas12a or Cas9 edited FK506[R] transformants. The number of independent transformants genotyped (x) is listed (*n* = x) collected from two independent transformations. The frequencies show the average outcome across replicates. **c** Sanger sequencing results from representative PCR positive editing using either Cas12a *FKBP12*-guide1 or Cas9 *FKBP12*-guide1. PAM and target sequences are highlighted with different colors as indicated. Black triangles indicate the nuclease sites for Cas12a and Cas9, respectively. Letters in blue boxes highlight microhomology at the repair junction. DNA insertions are highlighted by bold letters or blue rectangle.

Insertion_mt indicates a mitochondrial (mt) DNA insertion. The type and size of each editing outcome is listed in the right panel. **d** The mutation frequency of PCR positive *Δfkbp12* mutants. The number of independent transformants genotyped (x) is listed (*n* = x) collected from two independent transformations. The frequencies show the average outcome across replicates. The FK506[R] transformants without detected mutation in the targeting region were not counted (*n* = 1). **e** Size of deletion and insertion for PCR positive *Δfkbp12* mutants. Deletions were considered negative values, while insertions treated as positive values. * Indicates a *p*-value ≤ 0.05 (Mann Whitney test, two-sided, *p*-value = 0.0395). Two insertion mutants were not included due to ambiguous insertion sequences. The number of independent transformants genotyped (x) is listed (*n* = x), collected from two independent transformations. Source data are provided as a Source Data file.

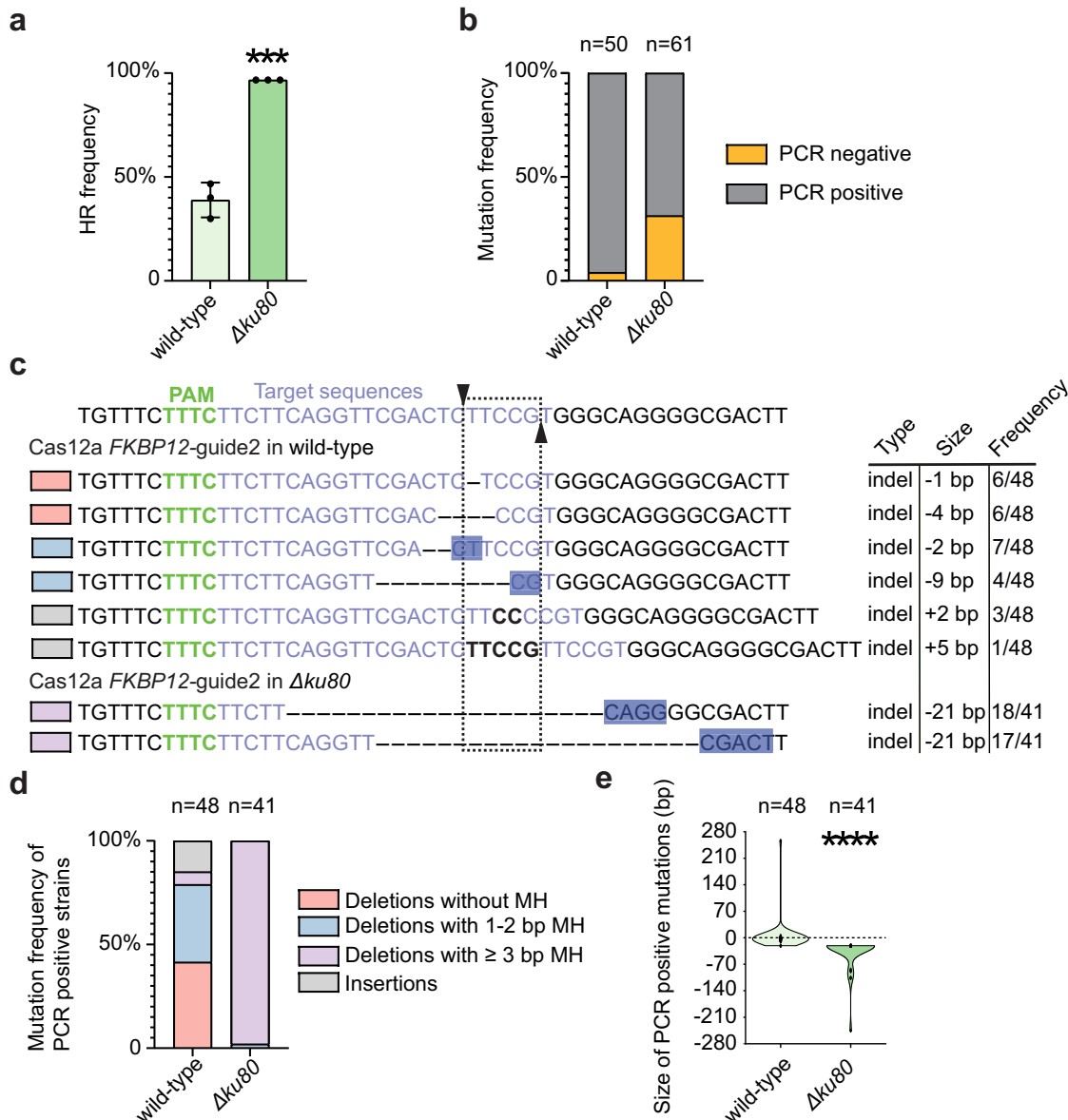

**Fig. 8 | The involvement of Ku80-independent DSB repair in large and microhomology-mediated DNA deletions. a** HR frequency in wild-type and *Δku80*. A total of 90 transformants were collected from *n* = 3 biological replicates. The HR frequency was determined per replicate, shown as a black dot. *** indicates a *p*-value ≤ 0.001 using unpaired *t*-test, two-sided, *p*-value = 0.0003. The plot shows a mean value 0.39 +/− 0.08389 for wild-type and mean value 0.97 +/− 0 for *Δku80*. **b** Mutation frequency of PCR negative and PCR positive outcomes in Cas12a-edited FK506ᴿ transformants in the background of wild-type or *Δku80*. The number of independent transformants genotyped (x) is listed (*n* = x) collected from two independent transformations for wild-type and three for *Δku80*. The frequencies show the average outcome across replicates. **c** Sanger sequencing for representative mutation outcomes from Cas12a *FKBP12*-guide2 in wild-type and *Δku80*. PAM and target sequences are highlighted with different colors as indicated. Black triangles indicate the nuclease site for Cas12a. Letters in blue boxes highlight microhomology at the repair junction. The type, size and frequency of each editing outcome is listed in the right panel. **d** The mutation frequency of PCR positive *Δfkbp12* mutants. The number of independent transformants genotyped (x) is listed (*n* = x) collected from two independent transformations for wild-type and three for *Δku80*. The frequencies show the average outcome across replicates. **e** Size of deletions and insertions from PCR positive *Δfkbp12* mutants. Deletions were considered negative values, while insertions treated as positive values. **** Indicates a *p*-value ≤ 0.0001 (Mann Whitney test, two-sided, *p*-value < 0.0001). The number of independent transformants genotyped (x) is listed (*n* = x) collected from two independent transformations for wild-type, and three independent transformations for *Δku80*. Source data are provided as a Source Data file.

At present, we cannot rule out that MMEJ participates in short deletions with short MH during wild-type DSB repair. Interestingly, in *M. oryzae* and related filamentous fungi, the homolog of PolΘ that is required for a-EJ is truncated and lacks the C-terminal DNA-polymerase domain (Supplementary Fig. 38). Yeast do not contain a homolog of this polymerase[65]. Our results provide a clear description of C-NHEJ-independent repair in *M. oryzae*, which we describe as MMEJ, but further mechanistic characterization of MMEJ in fungi is needed.

## Discussion

CRISPR-based genome engineering has accelerated functional genomic studies by providing a flexible platform to rapidly modify DNA and probe basic cellular and molecular biology[1,9]. Here we describe the development of an efficient and robust approach to modify specific loci in fungi using the Cas12a nuclease, delivered as an RNP. The use of RNPs for genome editing has the advantage of not requiring the integration and continual expression of the CRISPR-Cas system[51,66,67]. This can overcome cytotoxicity, a reported attribute of continual

expression of Cas nucleases in some systems including fungi, and is especially helpful for asexually reproducing fungi, where the CRISPR-Cas DNA cannot easily be removed through crossing[50,68]. The strains generated in this study were made with a high success-rate, contain heritable mutations, and lack coding sequence of the CRISPR-Cas platform. We anticipate the described approach to deliver Cas12a RNPs to fungal protoplasts will be fungal species-agnostic and provides a rapid approach to generate gene disruption mutations, especially for recalcitrant loci (e.g., pathogen effectors)[69,70].

Our results highlight an unresolved question at the interface of genome evolution and genome biology, which is how does hierarchy and crosstalk between endogenous DNA DSB repair pathways influence genome variation? Using a combination of PCR, Sanger sequencing and long-read sequencing-based assemblies, we show at base pair resolution how Cas12a-induced DNA DSBs can be variably repaired to generate a spectrum of DNA mutations. We observed INDELs, simple donor DNA insertions, large concatemer DNA insertions, large genomic deletions, and deletion plus insertion events, some of which resulted in drastic mutations at the targeted loci. It should be noted that error-free DNA repair may have also occurred at Cas12a-induced DSBs, but our approach was not able to select or quantify such events. The DNA repair mutation profiles, including those from the Δku80 mutant, show that at least three separate DNA repair pathways are active in M. oryzae. For C-NHEJ, we observed small INDEL mutations that contain no or short MH at their repair junctions, or the insertion of random sequence (i.e., no cis-template) or tandem sequences, which are well characterized signatures of this DSB repair dependent on Ku70-Ku80 and Lig4[20,35,41,71]. We also found that these repair signatures were lost or diminished when editing the Δku80 strain. The evidence supporting a functional MMEJ pathway is the occurrence of larger DNA deletions flanked by ≥3 bp microhomology that made up the majority of DNA mutations recovered in the Δku80 background. These DNA mutation patterns are signatures of MMEJ, and similar to those reported using Cas9 editing in mice[72]. However, details such as the proteins and repair outcomes of MMEJ are not well characterized in filamentous fungi[33]. The third likely pathway observed in our results is SSA, characterized by large deletions greater than 15 kb that are resolved between flanking repetitive DNA sequences. This occurred at both the BUF1 and BAS4 loci between distantly located repetitive DNA, similar to large deletions between repetitive sequences reported in the protozoan parasite Leishmania[73]. An alternative mechanism, such as loop mediated deletion during HR, is also possible[74]. Results from editing in the Δku80 background show that the large deletions are inhibited by Ku80. It is difficult to know from our experiments which DNA repair pathway was responsible for exogenous donor DNA integration. We frequently observed microhomology between the genome and integrated DNA. Additionally, the genomic site of DNA integration was often not exactly the same between integration events, a potential sign of DNA end resection prior to DNA integration. Both of these could indicate that the MMEJ pathway was used for donor DNA integration. Also, the majority of donor DNA integrations did not contain non-templated sequences at the junction, which has been reported for C-NHEJ knock-in insertions[27]. However, our experiments to integrate the HYG resistance coding sequence into the BUF1 locus in the Δku80 background was unsuccessful. This is despite the sequence being frequently integrated under the same conditions in the wild-type background, which suggests that Ku80 and C-NHEJ are required for the observed exogenous donor DNA integration. Future research is needed to resolve which pathway is responsible for donor DNA integration.

Along with demonstrating that activity of multiple DNA DSB repair pathways in M. oryzae, we observed that these pathways were used at unequal frequencies across the tested loci (i.e., locus-dependent frequencies). For example, editing the BUF1 and BAS4 loci resulted in large DNA deletions more often than the other three tested loci. Our observations that this can be mediated by flanking repetitive DNA, indicative of the SSA pathway, points to regional sequence factors on the scale of $10^3$ to $10^4$ kb that may influence the occurrence of SSA. Also at BAS4, we observed two independent cases of chromosome arm loss, suggesting that subtelomeric distribution might affect DNA DSB outcomes. The occurrence of other more complex insertion and deletion events also occurred more frequently at the BUF1 and BAS4 loci, although which repair pathway was responsible for these outcomes is not clear. Physical and genomic features such as chromatin structure, chromosome location, repetitive elements and the cell cycle have been reported to affect the outcome of DNA repair pathway choice in other model systems[15,19,33,75,76]. We interpret our findings to represent variation between repair pathway preference in the M. oryzae genome, and to suggest a locus-specific hierarchy. Intriguingly, locus-specific DSB repair outcomes have also been observed in the yeast Komagataella phaffii[77].

It is unlikely the observations reported here are specific to M. oryzae or our experimental setup. We found dramatic DNA alterations under multiple experimental designs (e.g., DNA donor with microhomology, no-homology or no DNA donor) and distinct nucleases (i.e., Cas12a and Cas9). Genome editing in other filamentous fungi, such as Sclerotinia sclerotiorum, Aspergillus fumigatus and Trichoderma reesei, reported abnormal genotyping results in which the target loci were larger than expected or PCR negative[78–80]. Further TAIL-PCR and Illumina sequencing suggested that vector sequences and many uncharacterized sequences were inserted in the target loci in the case of S. sclerotiorum[78]. Those experiments used the Cas9 effector, DNA-based delivery systems and different fungi, further suggesting our results are not specific to Cas12a, RNP delivery or M. oryzae[78,79]. Detailed studies in mice cell lines also suggested that large and complex DNA mutations are not dependent on transposition or delivery method and antibiotic selection[81]. These results suggest that our observations are not dependent on the use of RNP based CRISPR-Cas delivery, but we did not directly test editing outcomes using a vector delivery system. Future studies determining how vector versus RNP editing influence DNA repair will be of interest, especially in light of reports of Cas cytotoxicity in some fungal species[82].

It should also be noted, large-scale on-target mutations created by CRISPR are easily missed without comprehensive genotyping or de novo assembly such as employed here, and are likely heavily under reported[33,83]. Also, our repair outcomes are different than off-target editing often discussed for CRISPR research. Off-target editing refers to Cas nucleases binding and cutting DNA at unintended loci in the genome due to sequence homology (i.e., low-fidelity editing). Our results suggest that on-target DNA mutations following CRISPR DSB induction are varied and complex, consistent with recent reports in mammalian systems reporting extensive on-target mutations, including large deletions, complex rearrangements, and plasmid insertions[33,81,84,85]. These results underscore the need to further understand how CRISPR-based genome engineering interacts with endogenous DNA repair mechanisms.

Editing results from multiple loci in M. oryzae suggests a hierarchy for DNA pathway choice, which has significant evolutionary implications. Preferential repair of DSBs by different DNA repair pathways could create biased DNA variation prior to selection[33]. There are numerous reports of compartmentalized genome evolution in filamentous pathogens, often referred to as two-speed genome evolution[86], and variation for DNA DSB repair could be a major driver of this phenomena[33]. For instance, detailed analysis of DNA translocations and inversions between strains of Verticillium dahliae, a soil-borne wilt causing pathogen, found that chromosome rearrangements co-localize with homologous sequence, often transposable element DNA, that could serve as templates for homology-based

repair, such as MMEJ or SSA[87]. In *M. oryzae*, characterization of multiple translocations of the *Avr-Pita* coding sequence to different genomic locations likley required DNA repair[88]. The presence of dispensable mini-chromosomes described in *M. oryzae*[33,48], that contain duplicated sequences from core chromosomes, could impact homology based repair and have an uneven impact on the genome. Relevant to this is the recent finding in *Arabidopsis thaliana* that epigenetic features of the genome were associated with observed mutation bias[89]. Likewise, in *V. dahliae* hyper-variable adaptive genomic regions are associated with a unique chromatin profile[90]. Recently, a genome scale characterization of DNA repair pathway activity in a human cell line reported C-NHEJ and TMEJ are not uniformly active across the genome[75]. The authors reported that C-NHEJ functions more frequently at open euchromatin, while TMEJ has a larger contribution to DSB repair at specific heterochromatin domains[75]. Collectively, these results suggest that variation for DNA repair, mediated by the epigenome, can result in the creation of biased DNA variation. In filamentous fungal pathogens, this could have a substantial impact on the types of genome variation created and, by extension, pathogen evolution.

## Methods

### Fungal strains and incubation condition
*M. oryzae* field isolates O-137 (China) and Guy11 (French Guyana) were used as wild-types in this study[47,53]. *Δbuf1* mutants CP641 (gained from O-137 through spontaneous mutation) and CP281 (derived from weeping lovegrass pathogen 4091-5-8 through UV-mutagenesis) were used as a control in identifying the buff phenotype (Valent, unpublished data). JH7#1 and #2 are FK506 resistant mutants caused by a mutation in the *FKBP12* gene. The fungal cultures were maintained under light at 25 °C on OTA to observe mycelial color change. For high-molecular-weight DNA extraction and protoplast preparation, related mycelial plugs for different strains from OTA were cultured in liquid CM at 28 °C, 120 rpm for 3–4 days as described before[91,92].

### In vitro crRNA synthesis and LbCas12a or SpCas9 RNP assembly
Oligos including T7 promoter (taatacgactcactatagg), LbCas12a direct repeat (taatttctactaagtgtagat), and 23-nt target sequences (Supplementary Table 7) were annealed and amplified to make the DNA template for in vitro RNA synthesis. HiScribe™ T7 High Yield RNA Synthesis Kit (New England BioLabs, catalog# E2040S) was used to make the crRNA/gRNA with the above prepared DNA template according to manufacturer's protocol. Cas9 sgRNA synthesis kit (New England BioLabs, catalog# E3322S) was used for the Cas9 sgRNA preparation. Monarch® RNA Cleanup Kit (New England BioLabs, catalog# T2050L) was used to purify synthesized gRNA after DNase I (RNase-free) treatment (New England BioLabs, catalog# M0303S). 5 μg purified LbCas12a (New England BioLabs, catalog# M0653T or Integrated DNA Technologies, catalog# 10007923) or SpCas9 (New England BioLabs, catalog# M0646M) were incubated with equal molar purified gRNA at 25 °C for 15 min for RNP assembly[91].

### DNA donor preparation
pFGL821 (hygromycin selection, a gift from Dr. Naweed Naqvi; Addgene plasmid # 58223), and pFGL921 (G418 selection)[92] were used as DNA templates for amplifying DNA donor with related primer pairs (Supplementary Table 7). Homologous sequences of SS flanked *HYG* DNA donor were amplified from O-137 genomic DNA and inserted into KpnI/XbaI and SalI/PstI sites in pFGL821 (Supplementary Table 7). Long-flanking *HYG* DNA donor against *FKBP12* and long-flanking *G418* DNA donor against *KU80* were constructed via split marker method[93]. Phusion® High-Fidelity DNA Polymerase (New England BioLabs, catalog# M0530L) was used for DNA donor amplification.

### Protoplast preparation and polyethylene glycol (PEG) mediated transformation
*M. oryzae* protoplast preparation was performed as described previously[91]. The fungal mycelium was filtered and dried through 2-layer $6_{1/2}$ inch Disks Non Gauze Milk Filter papers, followed by addition of lysing solution (10 mg/mL Lysing Enzymes from *Trichoderma harzianum*, Sigma, catalog# L1412-10G, dissolved in 0.7 M NaCl solution) and digestion at 30 °C with 70–80 rpm for 2–3 h in the dark. After washing with 1xSTC (20% w/v Sucrose, 50 mM Tris-Hcl pH = 8.0, 50 mM $CaCl_2$ dissolved in water), the concentration of released protoplasts was adjusted to $8 \times 10^6 - 5 \times 10^7$ protoplasts/mL for transformation[91].

For protoplast transformation, RNP complexes (5 μg LbCas12a or SpCas9 protein complexed with equal molar amount of crRNA/gRNA) and/or 3 μg DNA donor were mixed with 200 μl concentrated protoplast at room temperature for 20–25 min. 1 mL 60% PEG solution (60% PEG4000, 20% w/v sucrose, 50 mM Tris-HCl pH = 8.0, 50 mM $CaCl_2$) was added to above mixture and incubated at room temperature for 20–25 min. This was followed by incubation with 5 mL TB3 liquid medium at 28 °C, 90 rpm, for 10–18 h. After overnight incubation, the fungal cultures were mixed with 50 mL molten (near 50–60 °C) TB3 solid medium containing 100 μg/mL hygromycin (Corning, catalog# 45000-806), 0.5 μg/mL FK506 (LC laboratories, catalog# NC0876958) or 300 μg/mL G418 (VWR, catalog# 97064-358). The fungal medium suspension was poured into a plate (150 × 15 mm), dried and then overlaid and cultured with another 50 mL molten TB3 solid medium plus 200 μg/mL hygromycin, 1 μg/mL FK506 or 600 μg/mL G418 in dark condition at 28 °C for 5–7 days. Potential fungal transformants were picked and sub-cultured on CM, OTA or RPA for further phenotyping and genotyping[91]. CM was supplemented with 1 μg/mL FK506, for testing the sensitivity to FK506.

### PCR genotyping
To test genotypes for the gene of interest, Q5® High-Fidelity DNA Polymerases (New England BioLabs, catalog# M0491) was used with 2 min extension time (technically up to 6 kb amplification) for the first-round genotyping with gene specific primer pairs (gene_F/R) (Supplementary Table 7). 5′ upstream, 3′ downstream regions and *ACTIN* were amplified with *Taq* DNA Polymerase (New England BioLabs, catalog# M0273) (Supplementary Table 7)[91]. Raw gel images are provided in a Source Data file.

### High-molecular-weight DNA extraction and library preparation for Nanopore sequencing
The DNA extractions for long-read nanopore sequencing were performed following the online protocol (https://www.protocols.io/view/high-quality-dna-from-fungi-for-long-read-sequenci-k6qczdw)[94]. g-Tube (Covaris, catalog# 520079) was used for shearing high-molecular-weight DNA into 20 kb fragments followed by purification with AMPure XP beads (Beckman, catalog# NC9959336). Nanopore sequencing library preparation followed the Native barcoding genomic DNA (with EXP-NBD104, EXPNBD114, and SQK-LSK109) protocol (https://community.nanoporetech.com/protocols/native-barcoding-genomic-dna/checklist_example.pdf). Eleven barcoded DNAs from independent Cas12a edited strains were sequenced in three nanopore MinION platforms for 72 h.

### Genome assembly and long-read mapping
Raw MinION fast5 files were transferred to fastq files by Guppy (version 3.4.4, https://nanoporetech.com/nanopore-sequencing-data-analysis) with the following parameters: --disable_pings --compress_fastq --flowcell FLO-MIN106 --kit SQK-LSK109. Adaptors were removed from basecalled reads by Porechop (version 0.2.4, https://github.com/rrwick/Porechop). Canu (version 1.9, 2.0 and 2.2.1, https://github.com/marbl/canu)[95] was used for de novo

genome assemblies with the following parameters: genomeSize = 45 m, minReadLength = 1500. Based on the flanking sequences, the contigs with *BUF1* flanking sequences (including #7, #8 assemblies from Fig. 3d) were merged manually to scaffold based on the alignment result, when the contigs containing *BUF1* are not intact. The raw long-read mapping was performed with minimap2 (version 2.17-r941)[96]. The mapping results were adjusted with samtools (version 1.9)[97] with following parameter: samtools view -q 60 to reduce the duplicated mapping and visualized with the Integrative Genomics Viewer (IGV) (version 2.5.0)[98]

### Synteny analysis
The synteny plots between sequenced mutants (i.e., *Δbuf1* and *Δbas4*) and wild-type (O-137) were generated with Easyfig (version 2.2.5)[99] by with the following parameter: Blast option_Min. length 30, Max. e Value 0.001. Each individual synteny plot was modified with Adobe Illustrator.

### Statistical analysis
Fisher's exact test was performed in RStudio (Version 1.2.5001) with function (fisher.test). Unpaired *t*-test and Mann Whitney test were carried out by using GraphPad Prism (version 8.01 and 9).

### Phylogenetic and domain analysis
The protein sequences of human Polq homologs were extracted from NCBI Genbank (https://www.ncbi.nlm.nih.gov/genbank/) and FungiDB (https://fungidb.org/fungidb/app/). The sequences used for analysis included MGG_15295 (*M. oryzae*), NCU07411 (*Neurospora crassa*), FGRAMPH1_01G23295 (*Fusarium graminearum*), NP_955452.3 (*Homo sapiens*), NP_084253.1 (*Mus musculus*), NP_524333.1 (*Drosophila melanogaster*), NP_498250.3 (*Caenorhabditis elegans*), AT4G32700.2 (*Arabidopsis thaliana*), XP_015619406 (*Oryza sativa*) and Pp3c5_12930V3.1 (*Physcomitrella patens*). The Neighbor-joining tree was made with MEGA X (version version 10.0.1)[100] with 1000 bootstrap value. TBtools (version 1.068)[101] was used for visualizing the domain structure gained from pfam (http://pfam.xfam.org/).

### Reporting summary
Further information on research design is available in the Nature Portfolio Reporting Summary linked to this article.

## Data availability
The de novo assemblies of Cas12a-edited strains from nanopore sequencing have been deposited with the accession number PRJNA753862 in the NCBI BioProject database (https://www.ncbi.nlm.nih.gov/bioproject/).The base called nanopore reads after adapter removal have also been deposited under the same BioProject accession number PRJNA753862 with the Short Read Archive (SRA) accession number SRR15459267, SRR15459268, SRR15459269, SRR15459270, SRR15459271, SRR15459272, SRR15459273, SRR15459274, SRR20662591, SRR20662592, and SRR20662593. Source data with raw gel images and raw data are provided as a Source Data file. Gene sequence with MGGnumber is from 70-15 MG8 genome annotation and can be found in FungiDB (https://fungidb.org/fungidb/app). The sequence of polq homologs can be found in NCBI Genbank (https://www.ncbi.nlm.nih.gov/genbank/). *M. oryzae* field isolates O-137, Guy11 and derived mutants within this study are available upon reasonable request from the corresponding author and may require permit. Source data are provided with this paper.

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

## Acknowledgements

The research was funded by the United State Department of Agriculture-National Institute of Food and Agriculture (USDA-NIFA) awards no. 2018-67013-28492 to D.E.C. and no. 2017-67013-26525 to B.V., and the National Science Foundation Division of Molecular and Cellular Biosciences –Systems and Synthetic Biology award no. 1936800 to D.E.C. The funders had no role in study design, data collection and analysis, decision to publish, or preparation of the manuscript. Contribution number 22-059-J from the Kansas Agricultural Experiment Station. The authors would like to thank Dr. Qiang Wang (Auburn University), Dr. Sanzhen Liu and Dr. Huakun Zheng (Kansas State University) for helpful discussion during the preparation of the manuscript. J.H. would like to thank Dr. Richard Todd (Kansas State University) for critical review prior to re-submission, Haolang Jiang (Fujian Agriculture and Forestry University) for helpful suggestions for fungal transformation, Ruiyun Zeng (Kansas State University) for assistant in data visualization and Melinda Dalby, Nathan Ryan and Dr. Velazhahan Rethinasamy, Joel Oberkrom (Kansas State University) for technical support. The authors thank computational support from the Beocat High-Performance Computing cluster at Kansas State University.

## Author contributions

D.E.C. and J.H. conceived and designed the project; J.H. performed the experiments with help from D.R. P.S. and T.S.; J.H. and W.Z. performed the nanopore sequencing. J.H. assembled and annotated the genomes in conjunction with D.E.C.; J.H., B.V. and D.E.C. analyzed the experiments; All authors contributed to writing the manuscript.

## Competing interests

The authors declare no competing interests.
