## [Peer Review File · Nature Communications]

CRISPR-Cas12a induced DNA double-strand breaks are repaired by multiple pathways with different mutation profiles in *Magnaporthe oryzae*REVIEWER COMMENTS

Reviewer #1 (Remarks to the Author):

This is a very well written manuscript which would be of broad interest to anybody using gene editing technology. There are however some issues with the work presented which I believe need to be addressed before the manuscript might be considered acceptable to publish. The study is to my mind rather preliminary in nature not least of all because it would appear most of the experiments shown have been only done one time and the number of events analysed is too small and there are therefore some doubts whether the conclusions drawn are very robust. Furthermore, the authors make very broad conclusions based on these small numbers, for example, one experiment reports 3 out of 36 as 8% of the population and another example 1 out of 15 is reported as approximately 6% of the population. These numbers are too small and experiments need to be repeated at least three times to know if this is robust/meaningful. The authors would need to repeat all the experiments at least three times so that they have looked at hundreds of transformants for each of the gene analysed in my opinion. Only then will there be an acceptable number of transformants and experimental repeats. It is not possible to conclude anything very much from an experiment where only one gene edited strain was obtained (Fig 5). In the discussion the authors make broad claims as to the significance of their findings in DNA repair in general and a major conclusion, which would be significant if supported by better data, is that the DNA repair might operate in different ways depending on the genome context. There is not sufficient support for this however given the small numbers of events analysed and the fact that the one gene they looked at in most detail is a known hotspot for mutation. Additionally, there is the fact that only one experiment was used for each particular treatment (it is important to look at reproducibility by repeating experiments >3x) and importantly they only looked at one type of break (Cas12a induced).

A further issue mentioned above with the study is the use of the BUF1 gene because as the authors acknowledge this is a hotspot for mutation. Although I appreciate that the authors have recognised this and have also looked at a handful of other genes, it is still the case that most of the manuscript focuses on this gene which you would imagine to be atypical in terms of the DNA repair that you observe. I do appreciate that they have used the appropriate controls in this case and that no edited strains were generated using donor only controls, nevertheless some of the experiments made only on the buf1- mutants such as the detailed investigation into the nature of the large deletions and insertions should be done on the mutants generated by targeting at least one of the other genes.

What I would propose the authors need to do in order to improve the manuscript is to repeat all of the experiments at least three times but that additional experiments are required to support the broad claims made in the discussion. It is important that they make use of the mutants deficient in NHEJ (or that they use inhibitors of NHEJ) to better understand what kind of repair mechanisms are operating to generate the types of mutations they have observed in. They can do this for all the experiments shown. The authors also could perform whole genome sequencing and/or analysis of the large insertions/deletions for one of the genes other than BUF1. Additionally, experiments where no PCR product is taken as diagnostic for a large deletion could be further investigated by finding regions where amplicons could be generated and then design of primers to amplify across the proposed deletion (no amplicon is weak evidence alone). The manuscript could be made more compact by showing the experiments that were done for BUF1 together in a figure with the same experiments performed on the other gene targets. In the case of the final section of results (the so called SS experiments) I wonder why the authors have chosen not to use the gene avrPi9 in this case. I recommend that the authors conduct a parallel set of experiments targeting the same genes but using the enzyme Cas9 in order to understand if the types of repair seen are dependent on the type of DSB generated.

Could the authors additionally design donors with and without microhomology and compare these in their ability to repair the breaks in the genes examined (using both Cas12a and Cas9 and in wildtype and ku80- strains) and perhaps then comment further on the possible involvement of NHEJ and

microhomology-mediated repair pathways?
The methods are sufficiently detailed.

Reviewer #2 (Remarks to the Author):

The research presented by the authors provides very interesting and new insights in the complexity of genome editing in the fungal host strain studied by these authors, thus providing a good recommendation to publish this work. In particular, the fact that the authors take a critical evaluation of their research, by including numerous controls for potential biases can be seen as a strong point of the manuscript.

However, a few topics should be addressed to further support acceptance of the MS

1. It would significantly strengthen the paper in the authors would include a subset of the experiments in a NHEJ mutant to demonstrate the role of this pathway in any of the DSB repair mechanisms suggested.

2. Another aspects of the results presented, which is left untouched is whether the results are influenced by the fact that an RNP based approach is used instead of a more commonly used approach based on Cas and guide expression. Could the observed results be an "artefact" of the use of in vitro synthesized RNP and potential overload of the repair systems. It would be good to give this some more thoughts in the discussion

3. The authors refer to the generality (species-agnostic) of the results they obtained for *Magnaporthe*. However, this is not substantiated by experimental proof. In the manuscript results on only one species and only one nuclease were presented, which both could present specificities towards the obtained results. In particular the fact that in *Magnaporthe* a significant proportion of the genome consists of repeated sequences should be considered in this respect.

4. In general the figures clearly present the results described in the text. However, the is not the case in Fig 3. Due to the size of the figure sections a-d are difficult to read. For section e it is unclear what can be taken from this.

Reviewer #3 (Remarks to the Author):

The authors set out to identify the repair mechanism(s) of DSB caused by Cas12a RNPs in *Magnaporthe oryzae*. By targeting the BUF locus to generate an easily identifiable non colored mutants for further characterization, based on the types of mutations generated, the authors hypothesize at least three DNA repair mechanisms are involved. The authors then target other loci of interest, and use another strain of the fungus to further some of their conclusions.

This reviewer believes there is a possibility of a strong bias present in the data, as only mutants that have a functional HYG resistant phenotype are scored. In the first two experiments concerning the BUF locus (with and without flanking sequence for the HYG cassette), strains where the Cas12a RNP generated a DSB and were repaired error-free to the WT sequence in the absence of integration of the HYG cassette would not be scored. If this was a significant (although a currently unknown) amount, this would be in direct contrast to one of the main conclusions of the manuscript where "error-prone pathways" are present.

Most of the experiments carried out relied on at least two independent events to take place, 1) uptake of the Cas12a and DSB and 2) uptake of the HYG fragment. While still unable to detect an error-free DSB repair, the Cas12a targeting the FKBP12 locus to generate FK506 resistant mutants would allow the authors to assess the DNA repair outcomes in the absence of adding exogenous DNA i.e. the HYG cassette.

While the Cas12a targeting the SS locus for HYG integration and second Cas12a targeting another locus of interest did try to address some of these issues, there is still concern. When using the SS

locus for HYG integration, it implies that the Cas12a RNP cleavage rate is similar between both loci in calculating the mutation frequency, but this is not the case and was even shown that sgRNA sites differ in cleavage frequency in this manuscript. For instance if the sgRNA used to target BUF is much more efficient than the one targeting the SS locus, it would not be surprising to find most of the HYG resistant (had cleavage at the SS locus) also have the BUF locus mutated. Alternatively, if the sgRNA targeting the FKBP12 locus is less efficient than the one targeting the SS locus, then you would expect most of the FKBP12 to remain WT. However, if the sgRNAs targeting the FKBP12 and SS loci do have about the same efficiency, then this would support that most of the DNA repair occurred error-free.

The authors imply that these studies have implications to other fungi (line 105 and in the discussion), however outside of the few cited studies in the discussion, it is unknown if the proposed DNA repair mechanisms function in the same manner in other fungi. Further experimental evidence for this should be provided in support of these statements.

Dear Reviewers,

Please find our responses in blue. We have completed many replications and new experiments, as suggested by the reviewers, and the manuscript has significantly improved. We thank the reviewers for their comments and suggestions.

REVIEWER COMMENTS

Reviewer #1 (Remarks to the Author):

This is a very well written manuscript which would be of broad interest to anybody using gene editing technology.

Authors: Thank you for the positive feedback.

There are however some issues with the work presented which I believe need to be addressed before the manuscript might be considered acceptable to publish. The study is to my mind rather preliminary in nature not least of all because it would appear most of the experiments shown have been only done one time and the number of events analysed is too small and there are therefore some doubts whether the conclusions drawn are very robust. Furthermore, the authors make very broad conclusions based on these small numbers, for example, one experiment reports 3 out of 36 as 8% of the population and another example 1 out of 15 is reported as approximately 6% of the population. These numbers are too small and experiments need to be repeated at least three times to know if this is robust/meaningful. The authors would need to repeat all the experiments at least three times so that they have looked at hundreds of transformants for each of the gene analysed in my opinion. Only then will there be an acceptable number of transformants and experimental repeats.

Authors: Thanks for your suggestions, we have independently replicated the requested fungal transformation and characterized the mutants. More specific details are below.

It is not possible to conclude anything very much from an experiment where only one gene edited strain was obtained (Fig 5).

Authors: For the experiment in question, we performed the fungal transformation on two independent occasions, and harvested a total of 41 hygromycin resistant strains. Among these 41 transformants, only one showed FK506 resistance. In our opinion, this means that the *FKBP12* targeting efficiency under the Second-Site editing scheme is low. We are not sure why, and it cannot be because of the guide, as this guide worked efficiently in other assays. There may be some biological meaning to this, but future experiments are needed. However, these results do not have a significant impact on our findings, and we are reporting our results as they were collected.

In the discussion the authors make broad claims as to the significance of their findings in DNA repair in general and a major conclusion, which would be significant if supported by better data, is that the DNA repair might operate in different ways depending on the genome context. There is not sufficient support for this however given the small numbers of events analysed and the fact that the one gene they looked at in most detail is a known hotspot for mutation. Additionally, there is the fact that only one experiment was used for each particular treatment (it is important to look at reproducibility by repeating experiments >3x) and importantly they only looked at one type of break (Cas12a induced).

Authors: We understand the claims are significant and we have performed the following experiments to better support our conclusions. 1). Fungal editing experiments for Fig1 and Fig2 used 2 guides with 3 replicates for O-137, and 2 guides with 2 replicates for Guy11. The use of a secondary genotype is a level of replication for the results, as is two guides per locus. This could be considered to have been repeated 10 independent times per experiment regarding *BUF1* editing outcomes. For experiments in Fig. 4, the 2 guides were each transformed 2 independent times. Again, the two guides are themselves controls for a locus, so each locus was checked through 4 independent experiments, each locus totaling at least 60 independent transformants per guide. For Fig.6, we used 1 guide and repeated the results for two independent replicates. For Fig.7, we used 1 guide, and repeated each Cas enzyme editing in 2 independent replications, totaling at least 50 independent transformants. For Fig. 8, we used 1 guide to perform two independent transformations of wild type and 3 independent

transformations of $\Delta ku80$. The importance of replication is to determine signal from noise and assess variation, and our experiments were replicated at the level of guides per gene, genotypes per gene, multiple loci per genome, and independent transformations per experiment. II). For another locus, *BAS4*, we nanopore sequenced 3 independent transformants to compare our results to the original analysis of the *BUF1* locus (Fig. 5). Long-read sequencing and assembly also showed how chromosome location of *BAS4* resulted in locus-dependent DNA repair outcomes. III). We compared the editing profiles between Cas12a and Cas9 at the *FKBP12* locus in the absence of donor DNA (Fig. 7). Here we show that the spectrum of DNA repair outcomes is not dependent on Cas12 or our original experimental design. We recovered a range of mutation types for Cas9 editing with direct *FKBP12* selection (Fig. 7). Together, these new experimental results further strengthen our conclusions, and we believe satisfy the reviewers concerns regarding bias and reproducibility. We understand that many more years and publications will be needed to fully test all the hypotheses related to our claims, but our work provides clear experimental results and claims that are timely and will be of interest.

A further issue mentioned above with the study is the use of the *BUF1* gene because as the authors acknowledge this is a hotspot for mutation. Although I appreciate that the authors have recognised this and have also looked at a handful of other genes, it is still the case that most of the manuscript focuses on this gene which you would imagine to be atypical in terms of the DNA repair that you observe. I do appreciate that they have used the appropriate controls in this case and that no edited strains were generated using donor only controls, nevertheless some of the experiments made only on the *buf1*- mutants such as the detailed investigation into the nature of the large deletions and insertions should be done on the mutants generated by targeting at least one of the other genes. Authors: As discussed above, we increased our experiments and have measured the editing efficiency and mutation profiles using two different guides for four different loci. We carefully genotyped all these strains using three different primer pairs to infer the results of DNA repair. Additionally, long-read DNA sequencing and *de-novo* assemblies were performed for three transformants for the *BAS4* effector coding locus (Fig 5). The results confirm our observations at the *BUF1* locus, and show dramatic DNA mutation effects as the result of DNA DSB repair, and how locus context in the genome matters. We would also like to draw attention to the reviewer's agreement about the *BUF1* locus being an 'unstable' locus. Our research demonstrates how *BUF1* and now *BAS4* can be unstable because of a single induced DNA DSB, and the results strongly support that various types of DNA editing outcomes result in a range of mutation profiles.

What I would propose the authors need to do in order to improve the manuscript is to repeat all of the experiments at least three times but that additional experiments are required to support the broad claims made in the discussion. It is important that they make use of the mutants deficient in NHEJ (or that they use inhibitors of NHEJ) to better understand what kind of repair mechanisms are operating to generate the types of mutations they have observed in. They can do this for all the experiments shown. The authors also could perform whole genome sequencing and/or analysis of the large insertions/deletions for one of the genes other than *BUF1*.

Authors: We have repeated the experiments, and also long-read sequenced and analyzed an additional locus, as the reviewer requested.

Additionally, we generated $\Delta ku80$ deletion mutants, which is a core component of the NHEJ pathway (FigS32). To confirm that our $\Delta ku80$ KO had the correct phenotype and was indeed deficient in NHEJ, we tested HR efficiency in $\Delta ku80$, and found highly efficient HR repair compared to WT (Fig. 8a and FigS33), consistent with previous reports (François Villalba et al., 2008 FGB and Yuuko Ninomiya et al., 2004 PNAS).

In order to test the role of *KU80* (i.e., NHEJ) in our observed DNA editing outcomes, we repeated our *BUF1* editing assay with no-homolog *HYG* donor in $\Delta ku80$. However, no hygromycin resistant transformants were recovered after 4 rounds of independent transformation. This result is different than editing in WT, and we conclude that NHEJ is required for no-homology donor DNA insertion.

Therefore, this experimental design and results were insufficient to understand the role of NHEJ and other repair pathways in our results.

To overcome this limitation, we edited the *FKBP12* locus, which allows for direct selection of the edited strains without the need for donor DNA integration. Therefore, we could edit the *FKBP12* locus in the wild-type and $\Delta ku80$ backgrounds and compare the mutation profiles in a donor-free assay (Fig. 8). We found that PCR negative mutants (i.e., large deletions) are significantly increased in $\Delta ku80$ (Fig. 8b). Comparing PCR positive mutants between the two backgrounds, we found larger DNA deletions with longer flanked microhomology when editing $\Delta ku80$ compared to WT (Fig. 8d and e). This result is consistent with the role of Ku80-Ku70 in DNA end protection, and provides clear genetic evidence that both NHEJ-dependent and -independent pathways (e.g., MMEJ and SSA) are activated in repairing Cas induced DSB in *M. oryzae* (Fig. 8). We can also conclude that the observed mutational profiles resulting from the different repair pathways is not the same.

Additionally, experiments where no PCR product is taken as diagnostic for a large deletion could be further investigated by finding regions where amplicons could be generated and then design of primers to amplify across the proposed deletion (no amplicon is weak evidence alone).

Authors: For every PCR negative strain in the paper, we performed two additional amplifications to check the presence of DNA upstream and downstream, approximately 500 bp from the original PCR primer pairs. This allowed us to assess if the negative PCR amplification was the result of a large DNA insertion (i.e., PCR negative for Cas targeting site, but PCR positive for upstream and downstream), or if a larger section of the locus had been deleted (i.e., one or both upstream and downstream products also failing to amplify). For both types of PCR negatives, it is nearly impossible to know how large a fragment of DNA was inserted or deleted, and therefore, it is not feasible to design PCR primers, nor test them on 100s of transformants. Additionally, it is not clear what information would be gained by having this data. All of the nanopore sequencing (Fig. 3 and 5) confirmed our interpretation of the PCR amplification results, along with Sanger sequencing of PCR positive strains. These results suggest our PCR genotyping and sequencing strategy is reliable.

The manuscript could be made more compact by showing the experiments that were done for BUF1 together in a figure with the same experiments performed on the other gene targets.

Authors: We considered this suggestion from the reviewer and believe our current presentation is the most interpretable for the reader. Combining figures could make the manuscript shorter, but we worry it would make understanding the different experiments more difficult. We believe the current layout strikes a balance between length and interpretability.

In the case of the final section of results (the so called SS experiments) I wonder why the authors have chosen not to use the gene *avrPi9* in this case.

Authors: Our reasoning was that *AvrPi9* had a nearly identical mutation profile to *FTR1*, and was similar to *FKBP12* (Fig. 4c), but had a lower editing efficiency. In an effort to cut down on some of the experiments, as this manuscript represents a huge amount of work, we had not included the *AvrPi9* locus for the SS experiment. We do not think including this locus in the SS experiments would impact our findings.

I recommend that the authors conduct a parallel set of experiments targeting the same genes but using the enzyme Cas9 in order to understand if the types of repair seen are dependent on the type of DSB generated.

Authors: Thanks for the suggestion, we compared the mutational profile between Cas12a and Cas9 in a donor-free editing scheme for the *FKBP12* locus (Fig. 7). We found similar repair outcomes (i.e., PCR negative, PCR positive including deletions without microhomology, deletions with 1-2 microhomology, deletions with ≥ 3 bp microhomology and insertions) when using either nuclease. Therefore, we can recover the full range of DNA mutations when using either Cas nuclease and this observation is not dependent on Cas12a. We note, the frequency and size of editing outcomes varied between Cas12a and Cas9, which may be of interest for further research (Fig. 7).

Could the authors additionally design donors with and without microhomology and compare these in their ability to repair the breaks in the genes examined (using both Cas12a and Cas9 and in wildtype and ku80- strains) and perhaps then comment further on the possible involvement of NHEJ and microhomology-mediated repair pathways?

Authors: Our experiments tested donor DNA with and without homology for *HYG* donor DNA and without homology for *G418* donor DNA. The results were similar for both types of assays, and appears to be donor DNA and selection-independent (Fig. S11). To understand the involvement of NHEJ-dependent and NHEJ-independent repair, we carried out new experiments in a newly created $\Delta ku80$ strain. We could not recover transformants when no-homology donor DNA was used, suggesting that NHEJ is required to integrate such no-homology DNA. We find that both Cas12a and Cas9 can result in a wide range of DNA mutational profiles following editing, and our observations are not dependent on the nuclease for DSB induction. Cas12a editing and repair resulted in a significantly different mutation profile for *FKBP12* editing in the wild type versus $\Delta ku80$ mutant. This result shows that NHEJ protects against larger scale mutations (i.e., more PCR negative in $\Delta ku80$), and that NHEJ-independent INDELS were larger and flanked by longer microhomology, which suggests they are the result of MMEJ. Taken together, our data show the involvement of both NHEJ-dependent and -independent pathways in repairing DSBs induced by Cas nucleases and that the profiles of these mutations are not the same at the tested loci.

The methods are sufficiently detailed.

Reviewer #2 (Remarks to the Author):

The research presented by the authors provides very interesting and new insights in the complexity of genome editing in the fungal host strain studied by these authors, thus providing a good recommendation to publish this work. In particular, the fact that the authors take a critical evaluation of their research, by including numerous controls for potential biases can be seen as a strong point of the manuscript.

Authors: Thanks for the positive feedback.

However, a few topics should be addressed to further support acceptance of the MS

1. It would significantly strengthen the paper if the authors would include a subset of the experiments in a NHEJ mutant to demonstrate the role of this pathway in any of the DSB repair mechanisms suggested.

Authors: Thanks for the suggestion, we generated the $\Delta ku80$ deletion mutants (FigS32), which is a core component of NHEJ pathway and showed the NHEJ deficient phenotype (Fig. 8a and FigS33). Therefore, we used this mutant to test the role of NHEJ in the DNA mutations observed.

We compared the mutational profiles between wild-type and $\Delta ku80$ for two loci. I) we edited the *BUF1* locus using the no-homology *HYG* DNA donor. Editing *BUF1* requires a donor DNA to be used in order to select transformants, as visible screening is not sufficient. However, we did not obtain any hygromycin resistant transformants in the $\Delta ku80$ strain, which suggested the requirement of NHEJ in no-homology DNA donor integration. We can make this conclusion as editing was routinely observed in the wild-type strain, and we could not recover any transformants from 4 independent rounds of transformation.

II) To test editing without the need for donor DNA, we edited an endogenous locus (i.e., *FKBP12*) that could be selected. We directly assessed the role of NHEJ-dependent and -independent repair without the confounding influence of donor DNA. Interestingly, there was a significant increase in PCR negative mutants (i.e., large deletions) found in $\Delta ku80$ compared to wild type, which suggested the involvement of NHEJ-independent DSB repair pathways (e.g., SSA or MMEJ) in generating those unexpected large deletions (Fig. 8b). Moreover, the PCR positive transformants from $\Delta ku80$ showed significantly larger INDELS and a longer microhomology profile (Fig. 8c and d). This is clear genetic

evidence of NHEJ-independent repair, and the results suggest MMEJ repair. Taken together, our new experiments in *Δku80* show NHEJ is highly active in *M. oryzae*, NHEJ inhibits other DNA repair pathways, other NHEJ-independent DNA DSB repair pathways are active in *M. oryzae*, and that the mutation profiles resulting from the different repair pathways are not the same.

2. Another aspects of the results presented, which is left untouched is whether the results are influenced by the fact that an RNP based approach is used instead of a more commonly used approach based on Cas and guide expression. Could the observed results be an "artefact" of the use of in vitro synthesized RNP and potential overload of the repair systems. It would be good to give this some more thoughts in the discussion

Authors: Thanks for the suggestion. It is not clear to us if the use of RNPs versus constitutive expression would be considered overloading the system. The use of RNPs may result in more Cas-guide molecules in the transformed cells, but after division, there are no Cas-guide complexes left. Stable expression of Cas/guide would result in continuous expression, and they are often under strong promoters, which could be argued to more likely overload the system. Never the less, we note that previous result from mouse cell line editing (Kosicki et al., 2018, Nature Biotechnology) found similar large DNA deletions and complex insertions following Cas9 editing. This research reported that mutational observations were not dependent on the delivery method or antibiotic selection. Therefore, we do not think our observations are the result of an artefact, given similar DNA mutations reported in a mouse model using Cas9 RNP and expression experiments. We included this information in the discussion.

3. The authors refer to the generality (species-agnostic) of the results they obtained for *Magnaporthe*. However, this is not substantiated by experimental proof. In the manuscript results on only one species and only one nuclease were presented, which both could present specificities towards the obtained results. In particular the fact that in *Magnaporthe* a significant proportion of the genome consists of repeated sequences should be considered in this respect.

Authors: We have modified our conclusion to be more specific to our results in *Magnaporthe*. Any discussion regarding the potential for these results to be more general includes specific references to published results. It is not our intention to make unsubstantiated claims, but it is our current hypothesis that it is unlikely *M. oryzae* is a significant outlier in terms of DNA repair, and there are observations in other studies that support this claim.

4. In general the figures clearly present the results described in the text. However, the is not the case in Fig 3. Due to the size of the figure sections a-d are difficult to read. For section e it is unclear what can be taken from this.

Authors: Thank you for the suggestions to make the figure easier to interpret. We have increased the size of the panels for Fig. 3a-d. We have modified some of the labels for section 3e and updated the text (lines 338-347) to improve the meaning of the result. The important take away from 3e is that mapping the reads to the reference genome indicated that these regions were deleted (i.e., there are no grey peaks representing read coverage), which is consistent with the results from the independent method of genome assembly.

Reviewer #3 (Remarks to the Author):

The authors set out to identify the repair mechanism(s) of DSB caused by Cas12a RNPs in *Magnaporthe oryzae*. By targeting the BUF locus to generate an easily identifiable non colored mutants for further characterization, based on the types of mutations generated, the authors hypothesize at least three DNA repair mechanisms are involved. The authors then target other loci of interest, and use another strain of the fungus to further some of their conclusions.

This reviewer believes there is a possibility of a strong bias present in the data, as only mutants that

have a functional HYG resistant phenotype are scored. In the first two experiments concerning the BUF locus (with and without flanking sequence for the HYG cassette), strains where the Cas12a RNP generated a DSB and were repaired error-free to the WT sequence in the absence of integration of the HYG cassette would not be scored. If this was a significant (although a currently unknown) amount, this would be in direct contrast to one of the main conclusions of the manuscript where "error-prone pathways" are present.

Authors: We have removed the term 'error-prone' from the manuscript. The reviewers correctly points out that we are not able to estimate or account for the number of DSBs that are repaired with high-fidelity (i.e. no mutation). This does not affect the conclusions of our manuscript or findings, which emphasize DNA mutation profiles across the genome and the role of different DSB repair pathways. Our emphasis is on the observable errors created during DSB repair. If error-free repair is taking place equally in the genome it is not that relevant, and if there is variation for error-free repair, it would further support our claims.

Most of the experiments carried out relied on at least two independent events to take place, 1) uptake of the Cas12a and DSB and 2) uptake of the HYG fragment. While still unable to detect an error-free DSB repair, the Cas12a targeting the FKBP12 locus to generate FK506 resistant mutants would allow the authors to assess the DNA repair outcomes in the absence of adding exogenous DNA i.e. the HYG cassette.

Authors: In order to reduce experimental biases caused by the donor DNA, we edited the *FKBP12* coding sequence and directly selected using FK506, as suggested by the reviewer (Fig7 and 8). As expected in the absence of donor DNA, more than 80% of the editing outcomes are now shorter deletions flanked with different lengths of microhomology. However, we still observe PCR negative genotypes, indicating that significant DNA mutations happen in the absence of donor DNA as a result of DNA DSB repair (Fig 8b, c and d).

While the Cas12a targeting the SS locus for HYG integration and second Cas12a targeting another locus of interest did try to address some of these issues, there is still concern. When using the SS locus for HYG integration, it implies that the Cas12a RNP cleavage rate is similar between both loci in calculating the mutation frequency, but this is not the case and was even shown that sgRNA sites differ in cleavage frequency in this manuscript. For instance if the sgRNA used to target BUF is much more efficient than the one targeting the SS locus, it would not be surprising to find most of the HYG resistant (had cleavage at the SS locus) also have the BUF locus mutated. Alternatively, if the sgRNA targeting the FKBP12 locus is less efficient than the one targeting the SS locus, then you would expect most of the FKBP12 to remain WT. However, if the sgRNAs targeting the FKBP12 and SS loci do have about the same efficiency, then this would support that most of the DNA repair occurred error-free.

Authors: Thank you for the thoughtful comments. As the reviewer mentioned, it is difficult to know the rates/kinetics between RNPs, and therefore certain conclusions from the SS experiment are confounded. For instance, the variation seen in Fig.6b may well be described by such rate differences for RNPs. But we do not make any strong claims regarding this, we are only reporting the primary-site editing frequency. Our main conclusion is drawn for the mutational profiles as shown in Fig.6c. We wanted to address the question, are mutation profiles different between different loci? The results support the hypothesis that mutational profiles are not the same between loci. Even if we only focus on *BAS4* and *FTR1*, which had a similar number of transformants and mutants recovered, the profile of mutations are not the same. As for *FKBP12*, the vastly different number of mutants recovered could be because of different RNP rates or because of error-free repair, both possibilities mentioned by the reviewer. However, even without knowing which is correct, neither possibility disproves our conclusion that repair profiles differ between loci. If the *FKBP12* locus does undergo more error-free repair, this would further support our claims. We cannot rule out differences in RNP rates, but we do not think we made substantial claims to this effect.

The authors imply that these studies have implications to other fungi (line 105 and in the discussion), however outside of the few cited studies in the discussion, it is unknown if the proposed DNA repair

mechanisms function in the same manner in other fungi. Further experimental evidence for this should be provided in support of these statements.

Authors: It was not our intention to overstate our findings. We do not think it likely that *M. oryzae* is an outlier for filamentous fungi, but we modified our discussion to be more circumspect regarding DNA repair in other fungi.

Editorial suggestions:

Please replace your bar graphs with plots that feature information about the distribution of the underlying data. All data points should be shown for plots with a sample size less than 10. For larger sample sizes, please consider box-and-whisker or violin plots as alternatives. Measures of centrality, dispersion and/or error bars should be plotted and described in the figure legend.

Authors: The stacked bar plots used in the manuscript describe classifications, not continuous variables. Box-whisker/violin plots are not appropriate for classification results. The results could be put in a table, but we find this less intuitive and informative for readers. We can make these tables, even on a per experiment basis if desired and add it to the already large collection of supplemental materials.

REVIEWERS' COMMENTS

Reviewer #2 (Remarks to the Author):

The authors have gone at great length to improve their publication by adding a considerable number of new data to support and finetune (in some cases downtune) their major conclusions. They have not only try to accomodate the reviewers comments but went above that, thus really improving the paper to become a very significant piece of work.

The only point I still have some concerns about is how they could accertain that their results are not at least partly related to using the RNP approach above the in vivo approach. The referral to the mammalian examples is not conclusve in my opinion. That there will be approach- gene-target- and species- specific differences regarding unscheduled editing profiles is in my mind clear and also demonstrated by the others on the gene-specific aspect. Another strong indication for this is that between fungi also classical editing approaches show species and gene-specific differences. Also the fact that in some fungi expression of cas9 even without the presence of guides severely reduces viability while in other species this is not the case supports this species specific aspect.

Finally I would encourage the others to include reference to a recent paper from a similar study in Pichia <https://www.mdpi.com/2309-608X/8/10/992> coming to similar kind of results

Reviewer #3 (Remarks to the Author):

In this revised manuscript, the authors have addressed my previous concerns. They have repeated and further refined many of the experiments and provide strong evidence that at least three DNA repair mechanisms are present in the fungus.

Dear Reviewers,

Please find our responses in blue. We thank the reviewers for their comments and suggestions.

REVIEWERS' COMMENTS

Reviewer #2 (Remarks to the Author):

The authors have gone at great length to improve their publication by adding a considerable number of new data to support and finetune (in some cases downtune) their major conclusions. They have not only try to accomodate the reviewers comments but went above that, thus really improving the paper to become a very significant piece of work.

The only point I still have some concerns about is how they could accertain that their results are not at least partly related to using the RNP approach above the in vivo approach. The referral to the mammalian examples is not conclusve in my opinion. That there will be approach- gene-target- and species- specific differences regarding unscheduled editing profiles is in my mind clear and also demonstrated by the others on the gene-specific aspect. Another strong indication for this is that between fungi also classical editing approaches show species and gene-specific differences. Also the fact that in some fungi expression of cas9 even without the presence of guides severely reduces viability while in other species this is not the case supports this species specific aspect.

Finally I would encourage the others to include reference to a recent paper from a similar study in *Pichia* <https://www.mdpi.com/2309-608X/8/10/992> coming to similar kind of results

Authors: Thank you for the positive feedback. We have revised the discussion about the potential effects on different approach-, gene-target and species-specific based on the suggestion (Line 623-627). The study from *Komagataella phaffii* also has been included (Line 609-610).

Reviewer #3 (Remarks to the Author):

In this revised manuscript, the authors have addressed my previous concerns. They have repeated and further refined many of the experiments and provide strong evidence that at least three DNA repair mechanisms are present in the fungus.

Authors: Thanks for your positive feedback and help.